# Geometry-Aware Decoding with Wasserstein-Regularized Truncation and Mass Penalties for Large Language Models

**Arash Gholami Davoodi** [1]   **Navid Rezazadeh** [2]   **Seyed Pouyan Mousavi Davoudi**   **Pouya Pezeshkpour** [3]

## Abstract

Large language models (LLMs) must balance diversity and creativity against logical coherence in open-ended generation. Existing truncation-based samplers are effective but largely heuristic, relying mainly on probability mass and entropy while ignoring semantic geometry of the token space. We present *Top-W*, a geometry-aware truncation rule that uses Wasserstein distance—defined over token-embedding geometry—to keep the cropped distribution close to the original, while explicitly balancing retained probability mass against the entropy of the kept set. Our theory yields a simple closed-form structure for the fixed-potential subset update: depending on the mass–entropy trade-off, the optimal crop either collapses to a single token or takes the form of a one-dimensional prefix that can be found efficiently with a linear scan. We implement Top-$W$ using efficient geometry-based potentials (nearest-set or $k$-NN) and pair it with an alternating decoding routine that keeps the standard truncation-and-sampling interface unchanged. Extensive experiments on four benchmarks (GSM8K, GPQA, AlpacaEval, and MT-Bench) across three instruction-tuned models show that **Top-$W$** consistently outperforms prior state-of-the-art decoding approaches achieving up to $33.7\%$ improvement. Moreover, we find that Top-$W$ not only improves accuracy-focused performance, but also boosts creativity under judge-based open-ended evaluation.[1]

[1]Carnegie Mellon University [2]University of California, Irvine [3]Megagon Labs. Correspondence to: Arash Gholami Davoodi <agholami@andrew.cmu.edu>.

*Proceedings of the $43^{rd}$ International Conference on Machine Learning*, Seoul, South Korea. PMLR 306, 2026. Copyright 2026 by the author(s).

[1]We release the code at: https://github.com/arashgholami/top-w-decoding

## 1. Introduction

Large language models (LLMs) have made autoregressive text generation a central interface for modern NLP systems (Vaswani et al., 2017; Brown et al., 2020; Grattafiori et al., 2024; Singh et al., 2025; Comanici et al., 2025). While model scaling and post-training alignment strongly affect capability (Ouyang et al., 2022; Rafailov et al., 2023; Guo et al., 2025), the *decoding algorithm* used at inference time remains a critical (and often under-emphasized) determinant of factuality, coherence, diversity, and perceived creativity. In practice, small changes in decoding (e.g., truncation rules, entropy constraints, or temperature schedules) can shift outputs from repetitive and bland to imaginative yet unstable, even for the same underlying model distribution (Holtzman et al., 2020; Hewitt et al., 2022; Meister et al., 2023; Nguyen et al., 2025).

Most widely used decoding schemes are probability-driven. Greedy decoding and beam search favor high-likelihood continuations, often improving local consistency but producing generic, low-diversity text (Vijayakumar et al., 2018). Stochastic sampling increases diversity, but can also reduce coherence and lead to degenerate behaviors (Holtzman et al., 2020). To better manage this trade-off, practitioners commonly apply truncation heuristics such as top-$k$ sampling (Fan et al., 2018), nucleus (top-$p$) sampling (Holtzman et al., 2020), and newer alternatives such as locally typical sampling (Meister et al., 2023) and Min-$p$ sampling (Nguyen et al., 2025). Beyond truncation, contrastive decoding trades off likelihood and degeneration by comparing competing continuations (Li et al., 2023; Chuang et al., 2024).

A complementary perspective is to view decoding as *distribution shaping*: at each step, we transform the model's next-token distribution into a truncated/renormalized distribution that better matches desired generation attributes. Recent work makes this view explicit by constraining distributional properties such as entropy. In particular, Top-$H$ decoding adapts creativity and coherence by enforcing a bounded-entropy principle over the next-token distribution (Baghaei Potraghloo et al., 2026). While entropy-based control provides a powerful knob for exploration–exploitation, it still treats tokens as unstructured categories. In contrast, natural language tokens live in an embedding space where

distances encode graded semantic similarity, suggesting that *geometry-aware* truncation could better preserve semantic continuity while still encouraging diverse continuations. Optimal transport (OT) offers a principled way to compare and regularize distributions while respecting such underlying geometry (Villani, 2009; Peyré and Cuturi, 2019).

We introduce **Top-$W$ decoding**, a geometry-aware truncation rule that augments probability-based candidate selection with a Wasserstein-inspired objective over the next-token set. At each decoding step, given the model distribution $p \in \Delta^{|\mathcal{V}|}$, Top-$W$ selects a *crop* $S \subseteq \mathcal{V}$ and samples from the renormalized distribution $q_S(i) = \frac{p_i}{\Gamma_S}\mathbf{1}\{i \in S\}$ with retained mass $\Gamma_S = \sum_{i \in S} p_i$. The crop is chosen by minimizing an interpretable objective $F_{\lambda,\beta}(S)$ (Eq. (5)) that trades off: (i) *faithfulness* via a geometry-aware transport penalty $W_1(p, q_S)$ under a token metric $d$ (computed from token embeddings), (ii) *creativity* via $\lambda H(q_S)$, and (iii) *mass control* via a $\beta$-term that discourages overly small $\Gamma_S$. Intuitively, Top-$W$ preserves high-probability mass while discouraging crops that concentrate probability on near-duplicate continuations in the same semantic neighborhood, yielding a controllable way to increase diversity without sacrificing coherence and complementing entropy-bounded decoding (Baghaei Potraghloo et al., 2026).

Computing exact $W_1$ inside decoding is infeasible at vocabulary scale, so we optimize a tight dual-based surrogate that replaces the transport term with nearest-set distances $\text{dist}(i, S) = \min_{j \in S} d(i, j)$ (Lemma 4.2). This yields simple, geometry-aware per-token scores and leads to an efficient alternating procedure: an $f$-*step* updates the potential (equivalently, the per-token geometric penalties/bonuses induced by the current crop), and an $S$-*step* updates the crop. For fixed $f$, we show the optimal crop is *prefix-form* after sorting tokens by the scalar score $i \mapsto f(i) + \lambda \log p_i$ (Theorem 3.4), so selecting $S$ reduces to scanning prefixes rather than searching over $2^{|\mathcal{V}|}$ subsets. Moreover, Top-$W$ provides a unifying view: by appropriate choices of the underlying metric, Top-$W$ reduces to familiar mass/entropy truncations such as Top-$k$ and Top-$H$ (Section 4.3). We evaluate Top-$W$ at fixed temperature on GSM8K, GPQA, AlpacaEval, and MT-Bench (plus LLM-as-a-judge creative writing) across 15 ($T$, model) settings with $T \in \{0.5, 0.7, 1.0, 1.5, 2.0\}$ and 3 LLMs. Across these settings, Top-$W$ consistently outperforms prior state-of-the-art decoding approaches: 13/15 (GSM8K), 12/15 (GPQA), 12/15 (AlpacaEval), and 8/15 (MT-Bench), delivering gains in both accuracy and judged creativity, while adding only modest runtime overhead (ms/token).

## 2. Preliminaries

We fix a vocabulary $V = \{1, \ldots, n\}$ and a ground metric $d : V \times V \to \mathbb{R}_{\geq 0}$. Let $E \in \mathbb{R}^{n \times m}$ be the model's input embedding matrix with token embeddings $\hat{e}_i \in \mathbb{R}^m$.

**Embedding-induced geometry.** Top-$W$ measures how much probability mass must be *moved* when cropping a distribution, via a Wasserstein-1 term that depends on a ground metric. We choose a metric derived from token embeddings because embeddings are trained so that geometric proximity correlates with functional similarity in the model: tokens that play similar semantic/syntactic roles tend to have nearby representations, and moving mass between such tokens should incur smaller transport cost than moving mass between unrelated tokens. This makes $W_1$ act as a *semantic* penalty rather than a purely combinatorial one, and encourages cropping rules that discard low-probability tokens while preferentially retaining mass in neighborhoods of plausible alternatives.

**Embedding-induced ground metric.** We equip the vocabulary $V = \{1, \ldots, n\}$ with an embedding-induced ground metric. Let $\hat{e}_i \in \mathbb{R}^m$ be the input embedding of token $i$ and let $\tilde{e}_i^{\text{white}}$ denote its diagonal-whitened, $\ell_2$-normalized version (Appendix A). We use the corresponding diagonal Mahalanobis distance

$$d(i, j) := \left\| \tilde{e}_i^{\text{white}} - \tilde{e}_j^{\text{white}} \right\|_2. \tag{1}$$

Diagonal whitening reduces anisotropy in embedding space (so high-variance directions do not dominate distances) while keeping computation cheap and numerically stable.

**Model distribution, cropping, and entropy.** Given a distribution $p \in \Delta^{n-1}$ over $V$ and any nonempty $S \subseteq V$, define the retained mass $\Gamma_S := \sum_{i \in S} p_i$, and the cropped, renormalized distribution

$$q_S(i) := \frac{p_i}{\Gamma_S} \mathbb{1}\{i \in S\}. \tag{2}$$

The Shannon entropy $H(\cdot)$ satisfies

$$H(q_S) = -\frac{1}{\Gamma_S} \sum_{i \in S} p_i \log p_i + \log \Gamma_S. \tag{3}$$

**Kantorovich–Rubinstein (KR) dual objects.** Let $\mathcal{F}$ be the set of discrete 1-Lipschitz potentials on $(V, d)$, so

$$W_1(P, Q) = \sup_{f \in \mathcal{F}} \left( \mathbb{E}_P[f] - \mathbb{E}_Q[f] \right). \tag{4}$$

## 3. Top-$W$ Decoding

This section turns *geometry-aware truncation* into a computable next-token rule. In §3.1, we define the Wasserstein–entropy–mass objective $F_{\lambda,\beta}(S)$ and prove an *exact factorization* (Lemma 3.1), which makes the roles of geometry, entropy, and retained mass explicit. Optimizing $F_{\lambda,\beta}(S)$

is challenging because computing $W_1$ exactly would require solving the KR dual in 4 (or equivalently the OT linear program) at each decoding step, which is infeasible at vocabulary scale (Peyré and Cuturi, 2019). Instead, we propose a lightweight *alternating* scheme: an $f$-*step* that refines a feasible potential $f$, and an $S$-*step* that, for fixed $f$, optimizes a tight lower-bound surrogate in closed form. In §3.2 we focus on the $S$-*step* and show that for any fixed feasible $f$, minimizing the surrogate over $S$ is equivalent to maximizing a simple set function $G_f(S)$ (Lemma 3.2), which depends on $S$ only through its retained mass $\Gamma_S$. This yields a sharp structural characterization of the optimizer (Theorem 3.4): the optimal crop is obtained by scanning prefixes of tokens sorted by $\phi_i$, giving an $O(n)$ update rather than an $O(2^n)$ search (over a vocabulary of size $n = |\mathcal{V}|$). Moreover, we show monotonicity of the selected retained mass in $\beta$ (Corollary 3.5), which directly motivates our logits-processor implementation.

### 3.1. Wasserstein–Entropy–Mass Objective and an Exact Factorization

At each decoding step, the model defines a next-token distribution $p$ over the vocabulary $V$. A truncation rule chooses a nonempty set $S \subseteq V$ and samples from the *cropped and renormalized* distribution

$$q_S(i) = \frac{p_i}{\Gamma_S} \mathbb{1}\{i \in S\}, \qquad \Gamma_S = \sum_{i \in S} p_i.$$

Choosing $S$ is inherently a tradeoff: we would like the cropped distribution to (i) stay faithful to $p$, (ii) avoid overly diffuse (high-entropy) supports that harm coherence, and (iii) not discard too much probability mass, since removing high-mass tokens tends to degrade quality.

We encode these desiderata with the following objective. For any nonempty $S \subseteq V$, define

$$F_{\lambda,\beta}(S) := W_1(p, q_S) + \lambda H(q_S) - \beta \log \Gamma_S, \quad (5)$$

with $\lambda \geq 0$ and $\beta \geq 0$.[2]

The three terms have a direct interpretation: (i) $W_1(p, q_S)$ penalizes distributional distortion (e.g., geometric distortion under an embedding-based ground cost) introduced by cropping—it is small when the removed probability mass lies near (in the ground metric) the mass that remains; (ii) $\lambda H(q_S)$ penalizes *diffuse* crops and favors sharper, lower-entropy supports; and (iii) $-\beta \log \Gamma_S$ is a *mass reward*

---

[2]**Scale invariance (metric calibration).** If the ground metric is rescaled by a constant factor, $d'(i, j) = \alpha\, d(i, j)$ with $\alpha > 0$, then $W_1^{d'}(P, Q) = \alpha W_1^d(P, Q)$ for all $P, Q$. Consequently, for any $S \neq \emptyset$,

$$F_{\alpha\lambda,\alpha\beta}^{d'}(S) = \alpha F_{\lambda,\beta}^d(S).$$

Thus rescaling the embedding metric can be absorbed by $(\lambda, \beta) \mapsto (\alpha\lambda, \alpha\beta)$ without changing the optimizer $S$.

that discourages discarding too much probability, with a diminishing-returns effect through the logarithm. We therefore *minimize $F_{\lambda,\beta}(S)$*: a good crop should simultaneously be geometrically close to $p$, low-entropy after renormalization, and high-mass.

A key identity makes the Wasserstein term especially interpretable by separating *how much* mass is removed from *how far* the removed mass is from the retained mass.

**Lemma 3.1** (Exact factorization). *Let $S \subseteq V$ with $\Gamma_S \in (0, 1)$ and define $S^c$ as its complement in $V$. Then*

$$W_1(p, q_S) = (1 - \Gamma_S)\, W_1\big(p(\cdot \mid S^c),\, p(\cdot \mid S)\big), \quad (6)$$

*where $p(\cdot \mid S)$ and $p(\cdot \mid S^c)$ are the conditional distributions of $p$ restricted to $S$ and $S^c$, respectively.*

$W_1(p, q_S)$ separates into (i) the amount of removed probability mass, $(1 - \Gamma_S)$, and (ii) the geometric distance between the removed and retained conditional distributions. For proof of Lemma 3.1, see Appendix B. By Lemma 3.1 and the entropy identity (3), minimizing (5) is equivalent to minimizing the expanded form

$$\min_S F_{\lambda,\beta}(S) \text{ where}$$

$$F_{\lambda,\beta}(S) = (1 - \Gamma_S)\, W_1\big(p(\cdot \mid S^c),\, p(\cdot \mid S)\big)$$
$$+ (\lambda - \beta) \log \Gamma_S - \frac{\lambda}{\Gamma_S} \sum_{i \in S} p_i \log p_i. \quad (7)$$

### 3.2. Dual Surrogate and an Exact $S$-Step for Fixed Potential

The next lemma states that, for fixed $f$, the $S$-step reduces to maximizing a simple set function depending on the retained mass $\Gamma_S$ and $\{\phi_i(f)\}_{i \in V}$ where *combined score* is defined as $\phi_i(f) \overset{\text{def}}{=} f_i + \lambda \log p_i$.

**Lemma 3.2** (Fixed-$f$ $S$-step as normalized score maximization). *Fix any feasible $f \in \mathcal{F}$ and any subset $S \subseteq V$, and let $\Gamma_S := \sum_{i \in S} p_i$. Then $F_{\lambda,\beta}(S)$ admits the lower bound*

$$F_{\lambda,\beta}(S) \geq C_f - G_f(S), \quad (8)$$

*where $C_f = \sum_i p_i f_i$ is independent of $S$, and*

$$G_f(S) := \frac{1}{\Gamma_S} \sum_{i \in S} p_i\, \phi_i(f) + (\beta - \lambda) \log \Gamma_S. \quad (9)$$

*Consequently, for fixed $f$, minimizing the surrogate bound is equivalent to*

$$\arg\min_{S \subseteq V} \big(C_f - G_f(S)\big) = \arg\max_{S \subseteq V} G_f(S). \quad (10)$$

The derivation is algebraic (expand $\mathbb{E}_{q_S}[f]$ and $H(q_S)$ under $q_S(i) = p_i \mathbf{1}\{i \in S\}/\Gamma_S$) and is deferred to Appendix B.1.

**Remark 3.3.** Lemma 3.2 turns the $S$-step (for fixed $f$) into maximizing a single, easy-to-evaluate score $G_f(S)$. Instead of searching over all subsets $S \subseteq V$ (which is exponential in the vocabulary size), the objective depends on $S$ only through the retained mass $\Gamma_S$ and a normalized (renormalized) average of the combined scores over the kept tokens. This is exactly the structure that lets us solve the $S$-step efficiently by a simple scan over candidates (Theorem 3.4), i.e., in $\mathcal{O}(n)$ time per decoding step (up to sorting), where $n := |V|$.

**Theorem 3.4** (Exact fixed-$f$ $S$-step: prefix regime vs. singleton regime)**.** *Fix $f \in \mathcal{F}$ and define the combined scores $\phi_i := f_i + \lambda \log p_i$. Sort indices so that $\phi_{(1)} \geq \phi_{(2)} \geq \cdots \geq \phi_{(n)}$, and let $S_k := \{(1), \ldots, (k)\}$ denote the top-$k$ prefix in this order. For any nonempty $S \subseteq [n]$ write $\Gamma(S) := \sum_{i \in S} p_i$ and $\Phi(S) := \sum_{i \in S} p_i \phi_i$, and define*

$$G_f(S) := \frac{\Phi(S)}{\Gamma(S)} + (\beta - \lambda) \log \Gamma(S). \qquad (11)$$

*Then:*

1. ***Prefix optimality when*** $\beta \geq \lambda$***.*** *If $c := \beta - \lambda \geq 0$, an optimizer of $G_f(S)$ exists among the prefixes:*

$$\max_{S \neq \emptyset} G_f(S) = \max_{k \in [n]} G_f(S_k), \qquad (12)$$

*so the exact $S$-step reduces to a one-dimensional scan over $k$.*

2. ***Singleton collapse when*** $\beta \leq \lambda$***.*** *If $c \leq 0$, there exists an optimal singleton set $S = \{i^\star\}$, i.e.,*

$$\max_{S \neq \emptyset} G_f(S) = \max_{i \in [n]} \left\{ \phi_i + c \log p_i \right\}. \qquad (13)$$

**Interpretation.** In the prefix regime $\beta \geq \lambda$, the fixed-$f$ optimum is always a *prefix* of the tokens sorted by $\phi_i$: there exists $k^\star$ such that $S^\star = \{(1), \ldots, (k^\star)\}$. Thus the $S$-step is not a combinatorial search over subsets, but a one-dimensional scan over prefix length $k$. If $\beta \leq \lambda$ (singleton regime), the optimum collapses to a *single token*: there exists $i^\star$ such that $S^\star = \{i^\star\}$, so the $S$-step reduces to selecting the best singleton according to (13).

**Proof idea (main body).** Fix the retained mass $m = \Gamma(S)$. The term $\Phi(S)/\Gamma(S)$ is a $p$-weighted average of $\{\phi_i\}_{i \in S}$, hence it is maximized (for a given mass budget) by taking the largest $\phi_i$'s—which yields a prefix $S_k$. When $c = \beta - \lambda \geq 0$, the extra term $c \log m$ is increasing in $m$, and the optimum occurs at one of the attainable prefix masses, so scanning $k$ suffices. When $c \leq 0$, the mass bonus becomes a mass *penalty*; combining the weighted-average bound with $\log \Gamma(S) \geq \log p_i$ for $i \in S$ shows a best singleton achieves the optimum. Full proof is provided in Appendix C.

**Corollary 3.5** (Monotonicity in $\beta$ (fixed $f$, prefix regime))**.** *Fix $f \in \mathcal{F}$ and assume the prefix regime $\beta \geq \lambda$. Let $k^\star(\beta)$ be any maximizer over prefixes in (12). Then the retained mass is nondecreasing in $\beta$: if $\beta_1 < \beta_2$ (with both $\beta_j \geq \lambda$), then: $\Gamma\left(S_{k^\star(\beta_1)}\right) \leq \Gamma\left(S_{k^\star(\beta_2)}\right)$. Equivalently, increasing $\beta$ cannot decrease the selected prefix size / retained mass.*

The proof of Corollary 3.5 is given in Appendix D.

# 4. Computing Potentials and the Implemented Alternating Decoder

Section 3.2 showed a key structural fact: for any *fixed* feasible potential $f \in \mathcal{F}$, the subset update $S \leftarrow \arg\max_{S \neq \emptyset} G_f(S)$ is *exactly* solvable by a one-dimensional prefix scan (Theorem 3.4). What remains is how to obtain a useful feasible $f$ during decoding.

In principle, one would like to use a KR-optimal dual potential for the current pair $(p, q_S)$, but this is too expensive to compute at each decoding step, see Appendix E. Our implemented decoder therefore uses a cheap geometry-aware construction of $f$ and alternates: *(i)* update $f$ from the current $S$; *(ii)* update $S$ exactly for that $f$.

## 4.1. A Simple Geometry-Anchored Feasible Potential

The fixed-$f$ analysis in Section 3.2 holds for *any* feasible potential $f \in \mathcal{F}$ (the set of discrete 1-Lipschitz functions on $(V, d)$). Rather than solving a costly OT dual to obtain $f$, we pick a *cheap* potential that (i) is guaranteed feasible, and (ii) injects geometry (so truncation can depend on *semantic proximity under $d$*, not only ordered probabilities as in Section 3.1) through distances to the current set $S$.

**Lemma 4.1** (Shift invariance of the fixed-$f$ $S$-step)**.** *Fix $f \in \mathcal{F}$ and define $f' := f + c\mathbf{1}$ for any constant $c \in \mathbb{R}$. Then the fixed-$f$ subset objective has the same maximizers under $f$ and $f'$.*

For proof of Lemma 4.1 see Appendix F. By Lemma 4.1, we may normalize $f$ without changing the resulting crop. We therefore *anchor* the potential on the current set:

$$f(j) = 0, \qquad \forall j \in S. \qquad (14)$$

**Lemma 4.2** (Extremal anchored Lipschitz envelopes)**.** *Let $S \subseteq V$ be nonempty and define the distance-to-set function $\text{dist}(i, S) := \min_{j \in S} d(i, j)$. Then $\text{dist}(\cdot, S) \in \mathcal{F}$ and $-\text{dist}(\cdot, S) \in \mathcal{F}$. Moreover, for any $f \in \mathcal{F}$ satisfying (14),*

$$-\text{dist}(i, S) \leq f(i) \leq \text{dist}(i, S), \qquad \forall i \in V. \quad (15)$$

*Equivalently, among all feasible anchored potentials, $\text{dist}(\cdot, S)$ is pointwise maximal and $-\text{dist}(\cdot, S)$ is pointwise minimal (hence* extremal*).*

For proof of Lemma 4.2, see Appendix G. Based on Lemma 4.2, we would like to choose the *most attractive* anchored potential, i.e., the pointwise *minimum* feasible extension under $f|_S \equiv 0$:

$$f_S(i) := -\operatorname{dist}(i, S), \qquad i \in V. \tag{16}$$

This choice has two practical benefits: (i) it is always feasible and costs only distance-to-set queries (no LP/OT solve), and (ii) it produces the strongest geometric bias *toward $S$* allowed by 1-Lipschitzness: tokens farther from $S$ receive a more negative score. With this potential, the combined surrogate score becomes

$$\phi_i(f_S) = f_S(i) + \lambda \log p_i = -\operatorname{dist}(i, S) + \lambda \log p_i,$$

which prefers tokens that are both *likely* (large $\log p_i$) and *geometrically close* to the current set $S$ (small $\operatorname{dist}(i, S)$). (Using $+\operatorname{dist}(i, S)$ would instead be a repulsive variant that discourages proximity to $S$.)

### 4.2. Alternating Decoder: Exact $S$-step Inside a Practical Loop

At a single decoding step, the alternating procedure maintains $S^{(0)}, S^{(1)}, \ldots$ and repeats:

1. **(f-step)** build a feasible potential $f^{(t)} \in \mathcal{F}$ from $S^{(t)}$ (e.g., $f^{(t)} = f_{S^{(t)}}$ in (16));

2. **(S-step)** update $S^{(t+1)}$ by *exactly* maximizing $G_{f^{(t)}}(S)$ via the prefix scan of Theorem 3.4.

A small, fixed number of alternations is used in practice.

**Candidate pool (`top_m`).** To avoid full-vocabulary work, Algorithm 1 restricts all computations to a candidate pool $C$ containing the `top_m` most probable tokens under $p$. A sufficient condition under which this restriction is *exact* for the fixed-$f$ $S$-step is stated and proved in Appendix I.

### 4.3. Uniform-metric reductions

To connect Top-$W$ to standard *probability-only* truncation rules, we consider the uniform $(0-1)$ token metric $d_u(i, j) = \mathbb{1}\{i \neq j\}$, under which transport ignores geometry and depends only on the *retained mass* $\Gamma_S$. In this regime, $W_1(p, q_S)$ collapses to a simple function of $\Gamma_S$, and our objective becomes a probability-only criterion (see Appendix H). This yields clean reductions: (i) with a cardinality budget $|S| \leq k$ and $\lambda = \beta = 0$, optimizing our objective recovers Top-$k$; (ii) with $\beta = 0$, our objective is the Lagrangian relaxation of the entropy-constrained mass maximization problem underlying Top-$H$.

---

**Algorithm 1** Top-$W$ alternating subset update

---

**Require:** logits $\ell \in \mathbb{R}^n$; selection temperature $T_{\mathrm{sel}} > 0$; metric $d$ on $V$ (token geometry); parameters $\lambda \geq 0$, $\beta \geq 0$; alternations $T_{\mathrm{alt}} \in \mathbb{N}$.
1: Form $p$ by $p_i \propto \exp(\ell_i/T_{\mathrm{sel}})$.
2: Initialize $S^{(0)}$ using any probability-only rule (details deferred).
3: **for** $t = 0$ **to** $T_{\mathrm{alt}} - 1$ **do**
4:     **(f-step)** $f_i^{(t)} \leftarrow -\operatorname{dist}(i, S^{(t)})$ for all $i$.
5:     **(S-step)** $\phi_i^{(t)} \leftarrow f_i^{(t)} + \lambda \log p_i$.
    Sort so $\phi_{(1)}^{(t)} \geq \cdots \geq \phi_{(n)}^{(t)}$ and form prefix sums
    $\Gamma_k \leftarrow \sum_{r=1}^{k} p_{(r)}, \quad \Phi_k \leftarrow \sum_{r=1}^{k} p_{(r)} \phi_{(r)}^{(t)}$.
    $J_k \leftarrow \frac{\Phi_k}{\Gamma_k} + (\beta - \lambda) \log(\Gamma_k)$.
    $k^\star \in \arg\max_{k \in [n]} J_k, \quad S^{(t+1)} \leftarrow \{(1), \ldots, (k^\star)\}$.
6:     **if** $S^{(t+1)} = S^{(t)}$ **then**
7:         **break**
8:     **end if**
9: **end for**
10: Mask logits: keep $\ell_i$ for $i \in S^{(t_{\mathrm{final}})}$ and set $-\infty$ otherwise.

---

### 4.4. Mass parameter $\beta$: practical guidance

The mass parameter $\beta$ sets how strongly the $S$-update is rewarded for retaining probability mass (via $-\beta \log \Gamma_S$) relative to the entropy term: when $\beta \leq \lambda$, the exact fixed-$f$ update can collapse to a singleton (a greedy, single-token crop), while for $\beta > \lambda$ it instead prefers a nontrivial prefix-style crop whose retained mass (and typically its size) increases as $\beta$ increases (Theorem 3.4). Practically, we therefore choose $\beta > \lambda$ to avoid repeated singleton collapse, then tune $\beta$ upward to reduce over-truncation (retain more mass; larger crops) or downward to increase sharpness (retain less mass; smaller crops), noting that pushing $\beta$ too close to $\lambda$ risks re-entering the degenerate regime.

## 5. Experimental Details

**Models and decoding methods** We evaluate three LLMs—LLaMA-3.1-8B-Instruct (Grattafiori et al., 2024), Qwen2.5-3B-Instruct (Yang et al., 2025), and Phi-3-Mini-4K-Instruct (Abdin et al., 2024)—and compare four sampling rules under identical temperature, prompts, and stopping criteria: Top-$p$ nucleus sampling (Holtzman et al., 2020), Min-$p$ sampling (Nguyen et al., 2025), Top-$H$ (bounded-entropy cropping) (Baghaei Potraghloo et al., 2026), and our geometry-aware cropping method (Top-$W$). All methods are implemented as logits processors and use identical prompts, max token budgets, and stopping rules.

**Benchmarks and primary metrics** We evaluate on both accuracy-centric and judge-based benchmarks: GSM8K

| Temp | Qwen2.5 3B | | | | LLaMA-3.1-8B-Inst. | | | | Phi-3-Mini | | | |
|---|---|---|---|---|---|---|---|---|---|---|---|---|
| | Min-$p$ | Top-$p$ | Top-$H$ | Top-$W$ | Min-$p$ | Top-$p$ | Top-$H$ | Top-$W$ | Min-$p$ | Top-$p$ | Top-$H$ | Top-$W$ |
| 0.5 | **75.89** | 74.83 | 74.90 | 75.44 | 75.36 | 75.62 | 77.18 | **77.26** | 84.53 | 85.06 | 85.47 | **85.90** |
| 0.7 | 74.53 | 75.70 | 74.40 | **75.82** | 73.16 | 73.19 | **77.63** | 77.15 | 83.62 | 84.76 | 84.69 | **85.48** |
| 1.0 | 72.40 | 71.27 | 75.97 | **75.99** | 48.90 | 67.93 | 76.35 | **76.72** | 81.96 | 81.35 | 83.24 | **85.92** |
| 1.5 | 66.79 | 55.57 | 72.55 | **75.18** | 58.00 | 23.81 | 70.51 | **75.74** | 77.10 | 67.25 | 77.86 | **85.32** |
| 2.0 | 49.43 | 9.10 | 55.57 | **75.13** | 13.72 | 2.65 | 39.35 | **73.09** | 60.88 | 7.73 | 60.20 | **84.63** |

*Table 1.* GSM8K accuracy (%) at $T \in \{0.5, 0.7, 1.0, 1.5, 2.0\}$ for Min-$p$, Top-$p$, Top-$H$, and Top-$W$. Aggregated over 3 runs (N=1319).

| Temp | Qwen2.5 3B | | | | LLaMA3.1-8B-Inst. | | | | Phi-3-Mini | | | |
|---|---|---|---|---|---|---|---|---|---|---|---|---|
| | Min-$p$ | Top-$p$ | Top-$H$ | Top-$W$ | Min-$p$ | Top-$p$ | Top-$H$ | Top-$W$ | Min-$p$ | Top-$p$ | Top-$H$ | Top-$W$ |
| 0.5 | 29.24 | **30.58** | 27.68 | **30.58** | **31.92** | 29.24 | 30.36 | 30.80 | 30.92 | 31.81 | **32.14** | **32.14** |
| 0.7 | 27.01 | 27.46 | 27.68 | **31.47** | 30.13 | 29.24 | **30.58** | 29.24 | 28.57 | 29.91 | 32.59 | **33.04** |
| 1.0 | 28.35 | 27.68 | 28.79 | **30.02** | 26.34 | **32.81** | 29.24 | 30.80 | 31.92 | 30.58 | **32.37** | **32.37** |
| 1.5 | 30.13 | 27.23 | 27.90 | **30.58** | 28.35 | 28.57 | 30.58 | **33.26** | 29.91 | 28.57 | 30.80 | **31.13** |
| 2.0 | 25.00 | 22.32 | 28.12 | **28.46** | 26.12 | 23.88 | 28.79 | **31.02** | 23.44 | 18.53 | 30.80 | **31.64** |

*Table 2.* GPQA accuracy at $T \in \{0.5, 0.7, 1.0, 1.5, 2.0\}$ for Min-$p$, Top-$p$, Top-$H$, and Top-$W$. Aggregated over 4 runs ($N = 448$).

(Cobbe et al., 2021) and GPQA (Rein et al., 2024) are scored by exact-match accuracy (%) from extracted final answers (GSM8K) or the chosen option (GPQA). For AlpacaEval, we report length-controlled win rates using the standard length-debiasing protocol (Dubois et al., 2024). For MT-Bench, we report the average judge score (1–10) following the canonical pipeline (Zheng et al., 2023). To probe quality axes beyond task accuracy, we also run a rubric-based LLM-as-Judge evaluation on three creative prompts: for each prompt and temperature, we generate one response per method, score with a single judge across five rubrics (M1–M5), aggregate to an overall score, and randomize method order to mitigate positional bias. Appendix M.1 contains the prompt templates and rubrics.

**Evaluation protocol**   All methods are evaluated at temperatures $T \in \{0.5, 0.7, 1.0, 1.5, 2.0\}$ with identical max-token budgets, prompts/format constraints, and answer-extraction rules; only the truncation/cropping rule differs. We run GSM8K and GPQA via `lm-eval-harness` with benchmark-standard prompting and extraction (Gao et al., 2021). For AlpacaEval, MT-Bench, and the rubric-based creative evaluation, we use GPT-4o as the judge and randomize candidate order per item (and per repeat, when applicable). More details including judge prompts is in Appendix M.1.

**Top-$W$ configuration**   Top-$W$ applies cropping to the temperature-scaled next-token distribution and is controlled by $\lambda$ (entropy penalty) and $\beta$ (log-mass weight). Unless stated otherwise, we fix $\lambda = 2.2$ and $\beta = 2.8$ (see §6.2 and Appendix L). In implementation, we use a nucleus-style warm start, a fixed `top_m`= 1200 candidate pool, and `alt_iters`= 3 alternating refinement steps; these settings are fixed across tasks unless stated otherwise.

## 6. Experiments

We evaluate whether **Top-$W$** consistently improves over Min-$p$, Top-$p$ and Top-$H$ under matched temperature, prompts, max-token budgets, and stopping criteria. We report exact-match accuracy on GSM8K and GPQA, and judge-based outcomes on AlpacaEval, MT-Bench, and a multi-axis creative-writing rubric. We additionally sweep $(\lambda, \beta)$ to probe sensitivity and the accuracy–diversity trade-off. For GSM8K and GPQA, we quote the Min-$p$, Top-$p$ and Top-$H$ accuracy results from Baghaei Potraghloo et al. (2026). For AlpacaEval and MT-Bench, absolute scores depend on the judge model/prompt; we therefore run Min-$p$, Top-$p$, Top-$H$ (using their provided implementation), and Top-$W$ under our fixed judge configuration and protocol (Appendix J). We use the authors' released implementation of Top-$H$ and reproduce Min-$p$/Top-$p$ under the same evaluation pipeline; for GSM8K and GPQA we additionally quote baseline numbers reported in Baghaei Potraghloo et al. (2026).

### 6.1. Main Results

**Reasoning benchmarks (GSM8K, GPQA).**   Tables 1 and 2 report exact-match accuracy across $T \in \{0.5, 0.7, 1.0, 1.5, 2.0\}$. Across models, Min-$p$ and Top-$p$ degrade sharply as $T$ increases, while Top-$H$ and Top-$W$ are substantially more stable. On GSM8K, Top-$W$ wins (or mutually wins) 13/15 ($T$, model) settings across three models (Qwen2.5-3B, LLaMA-3.1-8B, Phi-3-Mini) and five temperatures $T \in \{0.5, 0.7, 1.0, 1.5, 2.0\}$; the improvement over Top-$H$ reaches up to 33.74% at $T$=2.0. Across all $T$, LLaMA-3.1-8B is the strongest absolute performer, while Qwen2.5-3B and Phi-3-Mini benefit more from Top-$W$'s

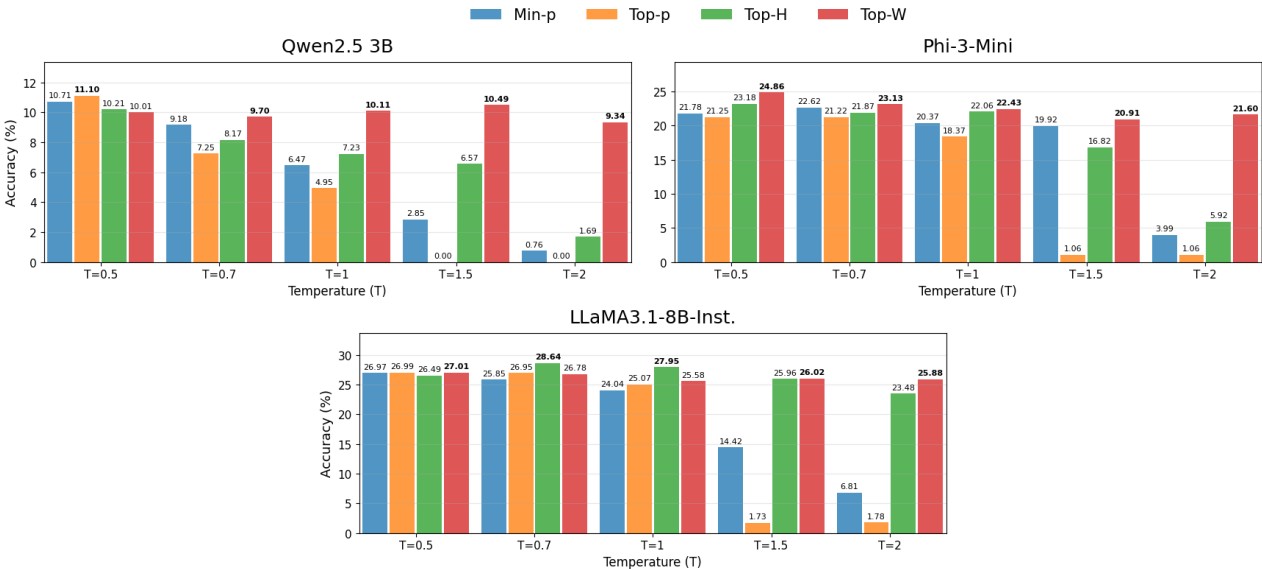

*Figure 1.* Alpaca accuracy across temperatures for Min-$p$, Top-$p$, Top-$H$, and Top-$W$ (aggregated over 4 runs). As we can see in the bar plot, Min-$p$, Top-$p$, Top-$H$ and Top-$W$ (our method) win in 0,1,2,12 tuples of $(T, model)$ out of 15 tuples, i.e., 5 temperatures × 3 models, respectively.

stability at higher $T$. GSM8K (multi-step arithmetic) is more sensitive to sampling noise, so probability-only truncators collapse faster than on GPQA, whereas Top-$W$/Top-$H$ preserve accuracy as $T$ rises. On GPQA (knowledge-heavy multiple-choice), accuracies are lower and tighter across methods, but Top-$W$ still delivers the most consistent gains, winning (or mutually winning) 12/15 $(T, model)$ *settings*: it dominates for all three models at $T \in \{1.5, 2.0\}$ and remains top or tied at $T{=}1.0$, while margins shrink at $T \in \{0.5, 0.7\}$ where baselines such as Min-$p$ (LLaMA-3.1-8B, $T{=}0.5$) and Top-$H$ (LLaMA-3.1-8B, $T{=}0.7$) occasionally win.

**Instruction-following and chat (AlpacaEval, MT-Bench).** Figures 1–2 summarize judge-based performance across temperatures and models. Top-$W$ achieves the strongest overall scores and remains robust as $T$ increases; it also wins the majority of $(T, model)$ tuples (e.g., 12/15 on AlpacaEval and 8/15 on MT-Bench in our reported aggregates). We observe larger method gaps on MT-Bench than AlpacaEval for some models, consistent with MT-Bench's multi-turn coherence and instruction retention being more brittle under increased stochasticity. AlpacaEval often rewards single-turn helpfulness/style, while MT-Bench more strongly penalizes drift and inconsistency, which amplifies the value of Top-$W$'s geometry-aware truncation at higher $T$.

**Creative writing rubric.** To probe fine-grained creative quality beyond pairwise win rates, we adopt an LLM-as-a-judge rubric (following Nguyen et al. (2025)) on a fixed set

of three open-ended storytelling prompts. The judge assigns 1–10 scores on five dimensions—*diversity* (novelty/uniqueness), *originality*, *narrative flow* (coherence), *emotional impact*, and *imagery*—and also selects an overall winner. We use GPT-4o as the judge, randomize the presentation order of methods to reduce positional bias, and report mean±std over repeats for each (LLM, $T$, prompt) configuration. For brevity, we move the full prompt- and LLM-specific tables to Appendix M; while absolute scores vary across prompts and backbones, the relative ordering is broadly consistent, with Top-$W$ achieving the strongest average rubric score and remaining robust at higher $T$ (Table 3). We additionally evaluate a higher-$\beta$ variant (Table 8 in Appendix), where increasing $\beta$ further improves rubric quality: across all 27 (LLM, $T$, prompt) triplets, Min-$p$, Top-$p$, and Top-$H$ win 6, 5, and 5 settings, respectively, while Top-$W$ wins 12.

### 6.2. Top-$W$ Ablations and Behavior

**Sensitivity to $\beta$ and the accuracy–creativity trade-off** Focusing on LLaMA-3.1-8B (because of its more divers behavior on Top-$W$) and GSM8K, we run a parameter sensitivity study over $\beta$ (and $\lambda$) to quantify how Top-$W$ transitions from conservative, accuracy-favoring behavior to more diverse generations (Figure 3). In particular, large $\beta$ can increase diversity in creative settings (as reflected by rubric-based judging), while potentially degrading performance on strict-answer reasoning tasks. As summarized by the sweep plots and the rubric-based judge tables (Tables 3 and 8), $\beta$ acts as a *control knob* on the exploration–conservatism axis: smaller $\beta$ tends to favor sharper truncation and higher

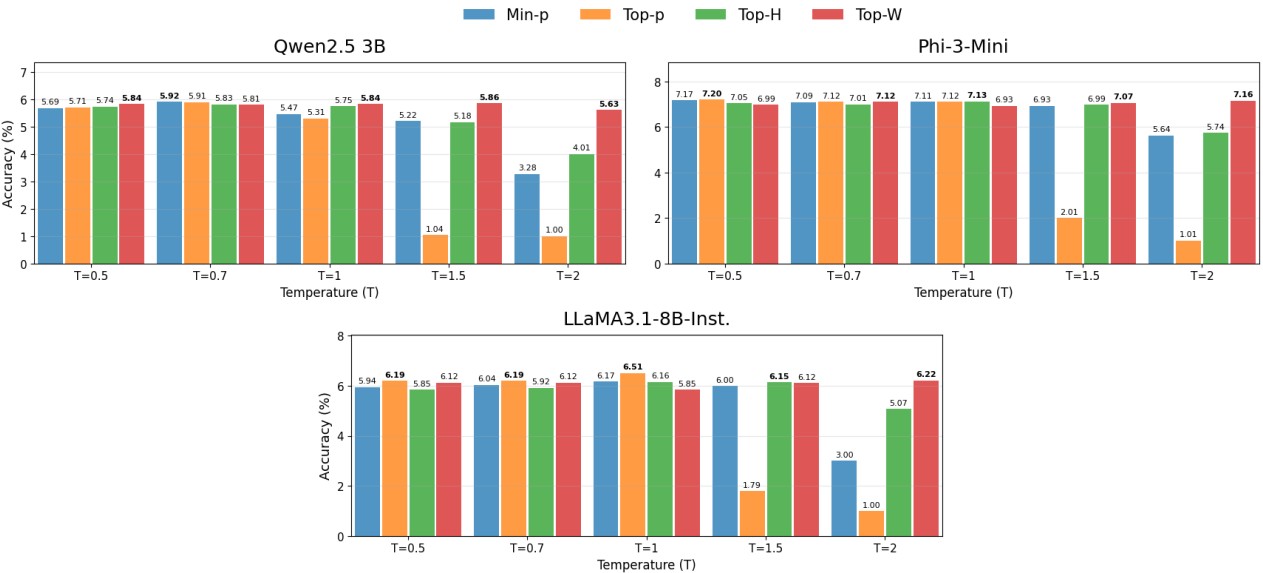

*Figure 2.* MT-Bench judge scores across temperatures for Min-$p$, Top-$p$, Top-$H$, and Top-$W$ (aggregated over 4 runs). As we can see in the bar plot, Min-$p$, Top-$p$, Top-$H$ and Top-$W$ (our method) win in 1,4,2,8 tuples of $(T, model)$ out of 15 tuples, i.e., 5 temperatures $\times$ 3 models, respectively.

| | | Qwen | | | | Llama | | | | Phi | | | |
|---|---|---|---|---|---|---|---|---|---|---|---|---|---|
| $T$ | Prompt | **Min-$p$** | **Top-$p$** | **Top-$H$** | **Top-$W$** | **Min-$p$** | **Top-$p$** | **Top-$H$** | **Top-$W$** | **Min-$p$** | **Top-$p$** | **Top-$H$** | **Top-$W$** |
| 1.0 | Prompt 1 | $6.64 \pm 0.88$ | $6.16 \pm 1.95$ | $6.45 \pm 0.57$ | $\mathbf{7.32 \pm 0.65}$ | $7.12 \pm 0.92$ | $7.12 \pm 0.63$ | $\mathbf{7.40 \pm 0.63}$ | $7.08 \pm 0.53$ | $6.96 \pm 0.59$ | $\mathbf{7.88 \pm 0.60}$ | $6.85 \pm 0.30$ | $6.88 \pm 0.74$ |
| 1.0 | Prompt 2 | $6.64 \pm 1.13$ | $7.00 \pm 0.62$ | $6.52 \pm 0.41$ | $\mathbf{7.12 \pm 0.90}$ | $7.16 \pm 0.15$ | $7.20 \pm 0.71$ | $\mathbf{7.36 \pm 0.34}$ | $6.95 \pm 0.71$ | $6.92 \pm 0.39$ | $\mathbf{7.72 \pm 0.93}$ | $7.28 \pm 1.03$ | $6.12 \pm 1.25$ |
| 1.0 | Prompt 3 | $6.64 \pm 0.74$ | $\mathbf{7.12 \pm 0.56}$ | $7.10 \pm 0.71$ | $5.56 \pm 0.87$ | $6.95 \pm 0.85$ | $\mathbf{7.80 \pm 0.51}$ | $7.44 \pm 0.23$ | $6.87 \pm 0.34$ | $7.28 \pm 0.63$ | $\mathbf{7.36 \pm 0.72}$ | $7.12 \pm 0.39$ | $7.12 \pm 0.79$ |
| 1.5 | Prompt 1 | $5.96 \pm 1.44$ | $1.08 \pm 0.10$ | $5.00 \pm 1.54$ | $\mathbf{7.16 \pm 0.75}$ | $\mathbf{7.88 \pm 0.39}$ | $3.84 \pm 2.03$ | $7.52 \pm 0.65$ | $7.68 \pm 0.60$ | $\mathbf{8.12 \pm 0.41}$ | $5.20 \pm 0.90$ | $7.80 \pm 0.47$ | $6.44 \pm 0.23$ |
| 1.5 | Prompt 2 | $\mathbf{7.60 \pm 0.55}$ | $1.12 \pm 0.10$ | $6.56 \pm 0.51$ | $6.00 \pm 1.76$ | $7.80 \pm 0.57$ | $2.16 \pm 0.62$ | $7.60 \pm 0.51$ | $7.45 \pm 0.36$ | $\mathbf{7.76 \pm 0.48}$ | $4.48 \pm 1.11$ | $7.60 \pm 0.68$ | $7.32 \pm 0.82$ |
| 1.5 | Prompt 3 | $6.92 \pm 0.93$ | $1.20 \pm 0.13$ | $\mathbf{7.04 \pm 0.75}$ | $6.88 \pm 0.77$ | $7.52 \pm 0.52$ | $1.64 \pm 0.29$ | $\mathbf{7.60 \pm 0.52}$ | $6.68 \pm 0.16$ | $7.12 \pm 0.59$ | $3.56 \pm 1.72$ | $7.08 \pm 0.60$ | $\mathbf{7.80 \pm 0.38}$ |
| 2.0 | Prompt 1 | $5.60 \pm 2.27$ | $2.00 \pm 1.90$ | $4.64 \pm 1.10$ | $\mathbf{7.16 \pm 0.73}$ | $6.92 \pm 0.59$ | $2.40 \pm 0.94$ | $6.20 \pm 0.58$ | $\mathbf{7.64 \pm 0.65}$ | $7.08 \pm 0.41$ | $2.20 \pm 1.22$ | $\mathbf{7.48 \pm 0.57}$ | $7.08 \pm 0.45$ |
| 2.0 | Prompt 2 | $\mathbf{6.72 \pm 1.04}$ | $2.04 \pm 1.98$ | $4.40 \pm 2.20$ | $6.15 \pm 0.65$ | $5.75 \pm 0.75$ | $2.20 \pm 1.72$ | $4.40 \pm 0.86$ | $\mathbf{7.64 \pm 0.57}$ | $\mathbf{7.76 \pm 0.86}$ | $2.08 \pm 1.00$ | $6.44 \pm 0.46$ | $7.12 \pm 0.59$ |
| 2.0 | Prompt 3 | $5.92 \pm 0.45$ | $1.20 \pm 0.25$ | $3.68 \pm 0.77$ | $\mathbf{7.60 \pm 0.64}$ | $4.80 \pm 0.20$ | $1.72 \pm 0.65$ | $4.04 \pm 1.04$ | $\mathbf{7.72 \pm 0.32}$ | $\mathbf{7.92 \pm 0.37}$ | $1.36 \pm 0.23$ | $6.64 \pm 1.37$ | $7.40 \pm 0.46$ |

*Table 3.* Average judge score (mean $\pm$ std over repeats) for each temperature $T$ and prompt. For each $(T, \text{prompt})$ and model, the best (highest mean) method is bolded; ties are bolded for all maxima. For details on each of the, Diversity, Originality, Narrative Flow, Emotional Impact check the Appendix. Across all 27 triplets, Min-$p$, top-$p$, and Top-$H$ win 8, 5, and 5 cases respectively, while Top-$W$ wins 9. This experiment is conducted under $\beta = 2.8$ with similar settings as other experiments.

accuracy on strict-answer tasks, while larger $\beta$ expands the viable support and can improve diversity- and style-sensitive judgments.

**Runtime and time complexity** Time complexity of the Top-$W$ logits processor is derived in Appendix K. Empirically, we report ms/tok for Top-$W$, Top-$H$, Top-$p$, and Min-$p$, and compute relative overhead vs. Top-$p$ at the same temperature. Across models and temperatures, Top-$W$ incurs only modest overhead from the geometric refinement as shown in Table 4 in Appendix K; in our measurements it is approximately **5.4% slower** on average than Top-$H$ / Top-$p$ / Min-$p$ under the same settings.

## 7. Related Work

Existing decoding heuristics offer strong trade-offs between quality, diversity, and speed. However, most truncation rules select supports using only probabilities and entropy, ignoring token-space geometry.

**Mass-based truncation and probability-only heuristics.** Popular stochastic decoders crop by probability mass (top-$k$, nucleus/top-$p$) and renormalize before sampling (Fan et al., 2018; Holtzman et al., 2020). Follow-ups tune truncation with typicality or online controllers—e.g., locally typical sampling, $\eta$-sampling, Mirostat, and Min-$p$—but still rely primarily on ordered probabilities and/or entropy rather than an explicit token geometry (Meister et al., 2023; Hewitt et al., 2022; Basu et al., 2021; Nguyen et al., 2025).

**Entropy-bounded decoding.** Entropy-control methods treat randomness as a direct constraint/knob; most closely, **Top-$H$** selects a truncated support whose renormalized distribution satisfies a bounded-entropy criterion, yielding a coherence–creativity control (Baghaei Potraghloo et al.,

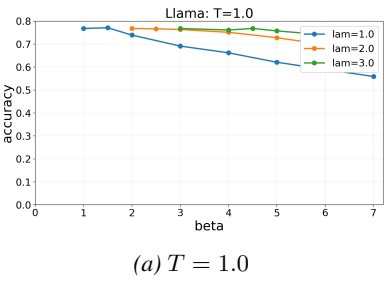 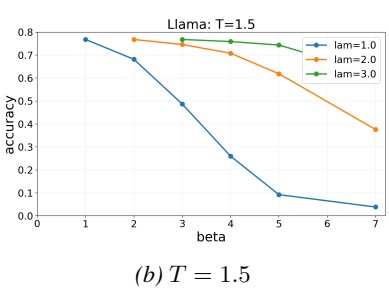 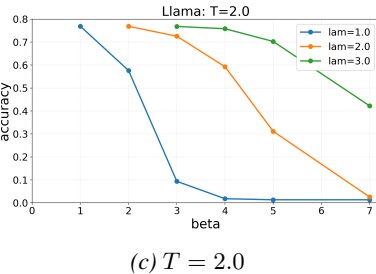

*(a)* $T = 1.0$      *(b)* $T = 1.5$      *(c)* $T = 2.0$

*Figure 3.* GSM8K accuracy sensitivity of Top-$W$ to $\beta$ for fixed $\lambda$s and LLaMA3.1-8B-Instruct at $T \in \{1.0, 1.5, 2.0\}$.

2026). Related entropy-based controllers (e.g., Mirostat and $\eta$-sampling) adapt truncation online via uncertainty/entropy proxies (Basu et al., 2021; Hewitt et al., 2022). Top-$W$ stays in this optimization-based truncation family, but adds a *transport* penalty under a token metric, making entropy-aware cropping sensitive to token-space structure.

**Geometry and optimal transport in NLP.** Embedding-induced OT has long been used to compare word-type distributions (e.g., Word Mover's Distance) (Kusner et al., 2015), and entropic OT provides scalable approximations (Cuturi, 2013). However, exact OT (and even heavy iterative approximations) is too expensive per decoding step at modern vocabulary sizes, motivating our restricted candidate pool and lightweight refinement.

**Logit shaping and acceleration.** A complementary line shapes logits using auxiliary signals or additional models (e.g., contrastive search, PPLM-style guidance, and GeDi-style discriminators) (Li et al., 2023; Dathathri et al., 2020; Krause et al., 2021), whereas Top-$W$ remains a single-model truncation sampler. Throughput-oriented acceleration such as speculative decoding is largely orthogonal and could, in principle, use Top-$W$ as the target distribution (Leviathan et al., 2023).

## 8. Conclusion

We proposed **Top-$W$**, a geometry-aware truncation rule that selects a cropped distribution by jointly trading off transport to the original next-token distribution (via $W_1$) and uncertainty (via an entropy term), yielding a principled generalization of widely-used truncation methods. We derived an efficient per-step implementation suitable for a logits processor, and showed how Top-$W$ connects to standard decoding rules under appropriate specializations.

Empirically, across a broad suite of evaluation settings spanning both accuracy-focused QA and judge-based open-ended generation, Top-$W$ consistently outperforms strong truncation baselines, winning in the large majority of cases (often nearly all), with the largest gains appearing in accuracy-oriented tasks and robust improvements also observed under LLM-judge protocols. Overall, these results suggest that incorporating embedding geometry into truncation decisions can reliably improve generation quality across diverse settings and evaluation regimes.

## Impact Statement

This work proposes an inference-time decoding method for large language models (LLMs) that leverages token-embedding geometry to improve generation quality. If adopted, it may yield positive societal benefits by enabling more coherent and diverse model outputs, potentially improving applications such as education, accessibility tools, programming assistance, and creative writing.

As with most advances that improve the fluency or controllability of text generation, our method could also increase the effectiveness of harmful uses of LLMs (e.g., misinformation, spam, harassment, or assisting other unsafe activities) by making outputs easier to produce or more persuasive. The method does not introduce new training data, does not modify model weights, and does not directly address (nor inherently worsen) underlying issues such as bias, toxicity, privacy leakage, or factual hallucination; however, changes in decoding can shift output distributions and therefore may affect the prevalence of such behaviors in practice. We recommend deploying geometry-aware decoding only in conjunction with established safety measures (e.g., content filtering, policy-tuned models, monitoring, and rate limits) and evaluating impacts on refusals, toxicity, bias, and hallucination rates in the target setting.

Overall, we view this contribution as a technical advance in large langue models. The broader impacts are largely those already associated with LLM deployment, with the main additional consideration being that improved decoding efficiency and output quality can amplify both beneficial and malicious downstream uses.

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

## A. Details of the embedding-induced metric

**Normalization and diagonal whitening.** Let $\hat{e}_i \in \mathbb{R}^m$ denote the input embedding for token $i \in V = \{1, \dots, n\}$ and define $\tilde{e}_i := \hat{e}_i / \|\hat{e}_i\|_2$. Compute the mean of normalized embeddings

$$\mu := \frac{1}{n} \sum_{i=1}^{n} \tilde{e}_i \in \mathbb{R}^m,$$

and the per-coordinate variance

$$\Phi_\ell := \frac{1}{n} \sum_{i=1}^{n} (\tilde{e}_{i\ell} - \mu_\ell)^2, \qquad \ell \in \{1, \dots, m\}.$$

For a small $\varepsilon > 0$, define the diagonal whitening matrix

$$D := \operatorname{diag}(s_1, \dots, s_m), \qquad s_\ell := (\Phi_\ell + \varepsilon)^{-1/2}.$$

The whitened embeddings are $\tilde{e}_i^{\text{white}} := D(\tilde{e}_i - \mu)$ and the resulting ground metric is

$$d(i, j) := \|\tilde{e}_i^{\text{white}} - \tilde{e}_j^{\text{white}}\|_2 = \sqrt{(\tilde{e}_i - \tilde{e}_j)^\top D^2 (\tilde{e}_i - \tilde{e}_j)}. \tag{17}$$

## B. Proof of Lemma 3.1

*Proof.* Fix any $f \in \mathcal{F}$. Using (2),

$$
\begin{aligned}
\mathbb{E}_p[f] - \mathbb{E}_{q_S}[f] &= \sum_i p_i f_i - \sum_{i \in S} \frac{p_i}{\Gamma_S} f_i \\
&= \sum_{i \in S^c} p_i f_i + \sum_{i \in S} p_i f_i - \frac{1}{\Gamma_S} \sum_{i \in S} p_i f_i \\
&= \sum_{i \in S^c} p_i f_i - \left( \frac{1 - \Gamma_S}{\Gamma_S} \right) \sum_{i \in S} p_i f_i \\
&= (1 - \Gamma_S) \Big( \mathbb{E}_{p(\cdot | S^c)}[f] - \mathbb{E}_{p(\cdot | S)}[f] \Big).
\end{aligned}
\tag{18}
$$

Taking the supremum over $f \in \mathcal{F}$ on both sides and applying the KR dual (4) to the pair $\big( p(\cdot \mid S^c), p(\cdot \mid S) \big)$ yields (6). $\qquad\square$

## B.1. Fixed-$f$ surrogate and proof of Lemma 3.2

We start from the KR dual

$$W_1(p, q_S) = \sup_{f \in \mathcal{F}} \left( \mathbb{E}_p[f] - \mathbb{E}_{q_S}[f] \right), \tag{19}$$

and fix any feasible $f \in \mathcal{F}$, yielding the lower bound

$$W_1(p, q_S) \geq \mathbb{E}_p[f] - \mathbb{E}_{q_S}[f]. \tag{20}$$

Recall the overall objective (definition in the main paper)

$$F_{\lambda,\beta}(S) := W_1(p, q_S) + \lambda H(q_S) - \beta \log \Gamma_S. \tag{21}$$

Combining (20) and (21) gives, for any feasible $f$,

$$F_{\lambda,\beta}(S) \geq \left( \mathbb{E}_p[f] - \mathbb{E}_{q_S}[f] \right) + \lambda H(q_S) - \beta \log \Gamma_S$$
$$= \mathcal{L}_f(S), \tag{22}$$

where $\mathcal{L}_f(S)$ is the fixed-$f$ surrogate.

**Expanding $\mathbb{E}_{q_S}[f]$ and $H(q_S)$.** By definition (2),

$$q_S(i) = \frac{p_i}{\Gamma_S} \mathbf{1}\{i \in S\}, \qquad \Gamma_S := \sum_{i \in S} p_i. \tag{23}$$

Hence,

$$\mathbb{E}_{q_S}[f] = \sum_i q_S(i) f_i = \frac{1}{\Gamma_S} \sum_{i \in S} p_i f_i. \tag{24}$$

For the entropy,

$$H(q_S) = -\sum_i q_S(i) \log q_S(i) = -\sum_{i \in S} \frac{p_i}{\Gamma_S} \log\left(\frac{p_i}{\Gamma_S}\right)$$
$$= -\frac{1}{\Gamma_S} \sum_{i \in S} p_i \log p_i + \frac{1}{\Gamma_S} \sum_{i \in S} p_i \log \Gamma_S$$
$$= -\frac{1}{\Gamma_S} \sum_{i \in S} p_i \log p_i + \log \Gamma_S. \tag{25}$$

**Closed form for $\mathcal{L}_f(S)$.** Substituting (24) and (25) into (22) yields

$$\mathcal{L}_f(S) = \sum_i p_i f_i - \frac{1}{\Gamma_S} \sum_{i \in S} p_i f_i - \frac{\lambda}{\Gamma_S} \sum_{i \in S} p_i \log p_i$$
$$+ (\lambda - \beta) \log \Gamma_S$$
$$= \sum_i p_i f_i + (\lambda - \beta) \log \Gamma_S$$
$$- \frac{1}{\Gamma_S} \sum_{i \in S} p_i (f_i + \lambda \log p_i). \tag{26}$$

**Rewriting as a normalized score maximization.** Using the definition of the combined score:

$$\phi_i(f) := f_i + \lambda \log p_i. \tag{27}$$

Then (26) becomes

$$\mathcal{L}_f(S) = \underbrace{\sum_i p_i f_i}_{=:C_f} + (\lambda - \beta) \log \Gamma_S - \frac{1}{\Gamma_S} \sum_{i \in S} p_i \phi_i(f)$$
$$= C_f - \left[ \frac{1}{\Gamma_S} \sum_{i \in S} p_i \phi_i(f) + (\beta - \lambda) \log \Gamma_S \right]$$
$$= C_f - G_f(S), \tag{28}$$

where $C_f$ is independent of $S$ and $G_f$ is exactly (9). Since $F_{\lambda,\beta}(S) \geq \mathcal{L}_f(S) = C_f - G_f(S)$, this proves the lower bound statement (8). Finally, because $C_f$ does not depend on $S$,

$$\arg\min_{S \subseteq V} \left( C_f - G_f(S) \right) = \arg\max_{S \subseteq V} G_f(S), \tag{29}$$

which is (10). $\square$

## C. Full proof of Theorem 3.4

*Proof.* Write $c := \beta - \lambda$. Assume $p_i > 0$ and $S \neq \emptyset$, so $\Gamma(S) > 0$.

**Part (a): prefix optimality for $c \geq 0$.** Fix any nonempty subset $S$ and let $m := \Gamma(S)$. Choose $k \in [n]$ such that

$$\Gamma_{k-1} < m \leq \Gamma_k, \tag{30}$$

where $\Gamma_k = \sum_{r=1}^k p_{(r)}$, and the convention $\Gamma_0 := 0$ and $\Phi_0 := 0$. Let $t := \phi_{(k)}$.

*Step 1: a discrete upper bound on $\Phi(S)$.* Decompose

$$\Phi(S) = \sum_{i \in S} p_i \phi_i = t \sum_{i \in S} p_i + \sum_{i \in S} p_i(\phi_i - t)$$
$$= tm + \sum_{i \in S} p_i(\phi_i - t). \tag{31}$$

Since $\phi_{(1)} \geq \cdots \geq \phi_{(n)}$, we have $\phi_i - t \leq 0$ for ranks $\geq k$ and $\phi_i - t \geq 0$ for ranks $\leq k-1$. Hence

$$\sum_{i \in S} p_i(\phi_i - t) \leq \sum_{i \in S \cap \{(1),\ldots,(k-1)\}} p_i(\phi_i - t)$$
$$\leq \sum_{r=1}^{k-1} p_{(r)}(\phi_{(r)} - t) = \Phi_{k-1} - t\Gamma_{k-1}. \tag{32}$$

Define the nonnegative constant

$$C_k := \Phi_{k-1} - t\Gamma_{k-1} = \sum_{r=1}^{k-1} p_{(r)}\big(\phi_{(r)} - \phi_{(k)}\big) \geq 0. \quad (33)$$

Combining (31) and (32) gives

$$\Phi(S) \leq tm + C_k. \quad (34)$$

Dividing by $m$ and adding $c \log m$ yields

$$G_f(S) = \frac{\Phi(S)}{m} + c\log m \leq t + \frac{C_k}{m} + c\log m =: F_k(m). \quad (35)$$

*Step 2: $F_k$ maximizes on $[\Gamma_{k-1}, \Gamma_k]$ at an endpoint when $c \geq 0$.* For $x \in [\Gamma_{k-1}, \Gamma_k]$, let $F_k(x) = t + C_k/x + c\log x$. Then

$$F'_k(x) = -\frac{C_k}{x^2} + \frac{c}{x} = \frac{-C_k + cx}{x^2}. \quad (36)$$

If $c \geq 0$, the numerator $-C_k + cx$ is nondecreasing in $x$, so $F'_k(x)$ crosses zero at most once, and if it crosses then it goes from negative to positive. Thus $F_k$ has no strict interior maximum on $[\Gamma_{k-1}, \Gamma_k]$, so

$$F_k(m) \leq \max\{F_k(\Gamma_{k-1}), F_k(\Gamma_k)\}. \quad (37)$$

Compute the endpoints using $C_k = \Phi_{k-1} - t\Gamma_{k-1}$:

$$F_k(\Gamma_{k-1}) = t + \frac{\Phi_{k-1} - t\Gamma_{k-1}}{\Gamma_{k-1}} + c\log(\Gamma_{k-1})$$

$$= \frac{\Phi_{k-1}}{\Gamma_{k-1}} + c\log(\Gamma_{k-1}), \quad (38)$$

$$F_k(\Gamma_k) = t + \frac{\Phi_{k-1} - t\Gamma_{k-1}}{\Gamma_k} + c\log(\Gamma_k)$$

$$= \frac{t\Gamma_k + \Phi_{k-1} - t\Gamma_{k-1}}{\Gamma_k} + c\log(\Gamma_k)$$

$$= \frac{\Phi_k}{\Gamma_k} + c\log(\Gamma_k). \quad (39)$$

Combining (35)–(39) yields

$$G_f(S) \leq \max_{j\in[n]}\left\{\frac{\Phi_j}{\Gamma_j} + c\log(\Gamma_j)\right\}. \quad (40)$$

Since each prefix $S_j$ is feasible, equality holds, and any maximizing $k^\star$ gives an optimal prefix $S_{k^\star}$. This proves (a).

**Part (b): singleton collapse for $c \leq 0$.** Let $S \neq \emptyset$ and write $m = \Gamma(S)$. Because $\Phi(S)/m$ is a weighted average of $\{\phi_i : i \in S\}$,

$$\frac{\Phi(S)}{m} \leq \max_{i\in S}\phi_i. \quad (41)$$

Also, for any $i \in S$ we have $m \geq p_i$, hence $\log m \geq \log p_i$. If $c \leq 0$, multiplying reverses the inequality:

$$c\log m \leq c\log(p_i), \qquad \forall i \in S. \quad (42)$$

Therefore

$$G_f(S) = \frac{\Phi(S)}{m} + c\log m \leq \max_{i\in S}\phi_i + c\log m$$

$$\leq \max_{i\in S}\{\phi_i + c\log(p_i)\} \leq \max_{i\in[n]}\{\phi_i + c\log(p_i)\}. \quad (43)$$

For a singleton $S = \{i\}$ we have $G_f(\{i\}) = \phi_i + c\log(p_i)$, so equality holds at any maximizing $i^\star$. This proves (b).

**Remark.** Since $\phi_i = f_i + \lambda\log p_i$ and $c = \beta - \lambda$, the singleton score can be written equivalently as $f_i + \beta\log p_i$; hence in the regime $\beta \leq \lambda$ the exact fixed-$f$ $S$-update is independent of $\lambda$. $\qquad\square$

## D. Proof of Monotonicity of the optimal prefix mass in $\beta$ for fixed $f$

*Proof.* Write $J_k(c)$ as an affine function of $c$:

$$J_k(c) = A_k + cB_k, \qquad A_k := \frac{\Phi_k}{\Gamma_k}, \quad B_k := \log(\Gamma_k). \quad (44)$$

Fix $c_1 < c_2$ and choose any $k_1 \in K(c_1)$ and $k_2 \in K(c_2)$. Optimality of $k_1$ at $c_1$ gives

$$A_{k_1} + c_1 B_{k_1} \geq A_{k_2} + c_1 B_{k_2}, \quad (45)$$

and optimality of $k_2$ at $c_2$ gives

$$A_{k_2} + c_2 B_{k_2} \geq A_{k_1} + c_2 B_{k_1}. \quad (46)$$

Adding (45) and (46) cancels the $A$-terms and yields

$$(c_2 - c_1)(B_{k_2} - B_{k_1}) \geq 0. \quad (47)$$

Since $c_2 - c_1 > 0$, we conclude $B_{k_2} \geq B_{k_1}$, i.e., $\log(\Gamma_{k_2}) \geq \log(\Gamma_{k_1})$, which is equivalent to $\Gamma_{k_2} \geq \Gamma_{k_1}$ because $\log$ is increasing on $(0, 1]$. Finally, if $p_i > 0$ for all $i$, then $\Gamma_k$ is strictly increasing in $k$, so $\Gamma_{k_2} \geq \Gamma_{k_1}$ implies $k_2 \geq k_1$. $\qquad\square$

## E. On Exact KR Solution

For a fixed crop $S$, the exact KR step solves the discrete dual linear program $\max_{f\in\mathbb{R}^n} \sum_{i=1}^n (p_i - q_S(i))f_i$ s.t. $|f_i - f_j| \leq d(i, j)$ $\forall i, j \in V$, which has $\Theta(n^2)$ Lipschitz constraints. Solving this discrete dual linear program at each decoding step (and potentially multiple times per token) is infeasible for modern vocabularies ($n \sim 10^5$). While entropic OT or graph-metric reductions can approximate dual potentials, they still require substantial iterative work and repeated distance access, making them impractical in a logits processor.

## F. Shift invariance of the fixed-$f$ subset update

**Theorem F.1** (Shift invariance of the fixed-$f$ $S$-step). *Fix $\lambda, \beta \geq 0$ and $p \in \Delta^{n-1}$. For any $f \in \mathbb{R}^n$ and any constant $a \in \mathbb{R}$, define $f' := f + a\mathbf{1}$. With*

$$\phi_i(f) := f_i + \lambda \log p_i, \qquad \Gamma_S := \sum_{i \in S} p_i, \qquad (48)$$

*and*

$$G_f(S) := \frac{1}{\Gamma_S} \sum_{i \in S} p_i \, \phi_i(f) + (\beta - \lambda) \log \Gamma_S, \qquad S \neq \emptyset,$$
$$(49)$$

*we have for every nonempty $S$,*

$$G_{f'}(S) = G_f(S) + a, \qquad (50)$$

*hence*

$$\arg\max_{S \neq \emptyset} G_{f'}(S) = \arg\max_{S \neq \emptyset} G_f(S). \qquad (51)$$

*In particular, the $\phi$-ordering and the optimal prefix/singleton returned by the exact scan are unchanged.*

*Proof.* For all $i$, $\phi_i(f') = f_i + a + \lambda \log p_i = \phi_i(f) + a$. Thus for any nonempty $S$,

$$\frac{1}{\Gamma_S} \sum_{i \in S} p_i \, \phi_i(f') = \frac{1}{\Gamma_S} \sum_{i \in S} p_i \big( \phi_i(f) + a \big)$$
$$= \frac{1}{\Gamma_S} \sum_{i \in S} p_i \, \phi_i(f) + a, \qquad (52)$$

since $\sum_{i \in S} p_i = \Gamma_S$. The term $(\beta - \lambda) \log \Gamma_S$ is unchanged, so $G_{f'}(S) = G_f(S) + a$. Adding a constant to all feasible objective values does not change the maximizers, and $\phi$-sorting is unchanged as well. $\square$

## G. Proof of Lemma 4.2

*Proof.* Fix $i \in V$ and any $j \in S$. Since $f \in \mathcal{F}$ and $f(j) = 0$, $|f(i)| = |f(i) - f(j)| \leq d(i, j)$. Minimizing over $j \in S$ yields (15). To show feasibility of $\mathrm{dist}(\cdot, S)$, fix $i, j \in V$ and let $s^\star \in S$ attain $\mathrm{dist}(i, S) = d(i, s^\star)$. By the triangle inequality, $\mathrm{dist}(i, S) = d(i, s^\star) \leq d(i, j) + d(j, s^\star) \leq d(i, j) + \mathrm{dist}(j, S)$, so $\mathrm{dist}(i, S) - \mathrm{dist}(j, S) \leq d(i, j)$; swapping $i, j$ gives the reverse inequality, hence $|\mathrm{dist}(i, S) - \mathrm{dist}(j, S)| \leq d(i, j)$ and $\mathrm{dist}(\cdot, S) \in \mathcal{F}$. Finally, $-\mathrm{dist}(\cdot, S) \in \mathcal{F}$ since negation preserves the Lipschitz constant. $\square$

## H. Uniform Metric and Reductions to Top-$H$, Top-$k$, Top-$p$, and Min-$p$

This appendix records probability-only *reductions*: under specific uniform geometry, several standard truncation rules

(Top-$k$, Top-$H$, and related heuristics) arise as special cases or natural relaxations of our objective. Concretely, replace the token metric by the uniform (0–1) metric $d_u(i, j) = \mathbb{1}\{i \neq j\}$. Under $d_u$, moving any amount of probability mass costs the same per unit regardless of token identity; thus transport is determined entirely by how much mass is discarded.

**Lemma H.1** (Uniform-metric closed form). *Under $d_u$,*

$$W_1(p, q_S) = 1 - \Gamma_S. \qquad (53)$$

Substituting (53) into (5) and using (3) gives

$$F_{\lambda, \beta}(S) = 1 - \Gamma_S + (\lambda - \beta) \log \Gamma_S - \frac{\lambda}{\Gamma_S} \sum_{i \in S} p_i \log p_i.$$
$$(54)$$

Equation (54) makes the mechanism explicit: *geometry disappears*, and $S$ influences the objective only through (i) its retained mass $\Gamma_S$ and (ii) the probability profile $\{p_i\}_{i \in S}$ (via the cropped entropy term).

### H.1. Reduction to Top-$k$

Top-$k$ is recovered by imposing a pure cardinality budget and optimizing retained mass. Impose $|S| \leq k$ and set $\lambda = \beta = 0$. Then (54) reduces to

$$F_{0,0}(S) = 1 - \Gamma_S. \qquad (55)$$

Minimizing $F_{0,0}$ is equivalent to maximizing $\Gamma_S$ subject to $|S| \leq k$, which is achieved by selecting the $k$ largest probabilities (Top-$k$).

### H.2. Reduction to Top-$H$ (entropy-constrained mass maximization)

Top-$H$ (Baghaei Potraghloo et al., 2026) studies entropy-constrained mass maximization:

$$\max_{S \subseteq V} \Gamma_S \quad \text{s.t.} \quad H(q_S) \leq \alpha \, H(p), \qquad (56)$$

for $\alpha \in (0, 1]$. Under $d_u$, setting $\beta = 0$ in (5) yields

$$F_{\lambda, 0}(S) = \big(1 - \Gamma_S\big) + \lambda \, H(q_S), \qquad (57)$$

which is the Lagrangian relaxation of (56) (up to the additive constant 1). Thus, in the uniform-metric regime, our objective matches the Top-$H$ tradeoff between *keeping mass* (via $1 - \Gamma_S$) and *controlling cropped entropy* (via $H(q_S)$).

## I. Candidate-pool exactness theorem (justifying Top-M restriction)

**Theorem I.1** (Candidate-pool exactness for the fixed-$f$ $S$-step). *Fix $f \in \mathcal{F}$ and let $c := \beta - \lambda \geq 0$. Define scores*

$\phi_i := f_i + \lambda \log p_i$ *and fix any deterministic tie-breaking so that $\phi_{(1)} \geq \phi_{(2)} \geq \cdots \geq \phi_{(n)}$ is a total order. For $k \in [n]$ define the full-vocabulary prefixes and their statistics*

$$S_k := \{(1), \ldots, (k)\}, \tag{58}$$

$$\Gamma_k := \sum_{r=1}^{k} p_{(r)}, \tag{59}$$

$$\Phi_k := \sum_{r=1}^{k} p_{(r)}\, \phi_{(r)}, \tag{60}$$

$$J_k := \frac{\Phi_k}{\Gamma_k} + c \log \Gamma_k. \tag{61}$$

*Let*

$$k^\star := \min \arg\max_{k \in [n]} J_k, \qquad S^\star := S_{k^\star}. \tag{62}$$

*Let $C \subseteq [n]$ be any candidate pool such that there exists an integer $L$ with $L \geq k^\star$ and*

$$\{(1), \ldots, (L)\} \subseteq C. \tag{63}$$

*Sort the elements of $C$ by decreasing $\phi$ using the* same *tie-breaking, and let $S_k^C$ be the top-k prefix within $C$ (for $k \leq |C|$). Define*

$$\Gamma_k^C := \sum_{i \in S_k^C} p_i, \tag{64}$$

$$\Phi_k^C := \sum_{i \in S_k^C} p_i\, \phi_i, \tag{65}$$

$$J_k^C := \frac{\Phi_k^C}{\Gamma_k^C} + c \log \Gamma_k^C. \tag{66}$$

*Let $k_C^\star := \min \arg\max_{k \in [|C|]} J_k^C$ and $S_C^\star := S_{k_C^\star}^C$.*

*Then $k_C^\star = k^\star$ and $S_C^\star = S^\star$.*

*Proof.* Because $C$ contains the global top-$L$ items in the $\phi$-order (63), sorting $C$ by $\phi$ yields

$$S_k^C = S_k, \qquad \Gamma_k^C = \Gamma_k, \qquad \Phi_k^C = \Phi_k, \qquad J_k^C = J_k, \tag{67}$$

$\forall k \in [L]$. In particular, since $k^\star \leq L$, we have $J_{k^\star}^C = J_{k^\star}$.

Next, every pool-prefix $S_k^C$ is a valid subset of $[n]$, hence it is feasible for the original maximization of $G_f(S)$. By Theorem 3.4(a), the global optimum value equals the best full prefix value:

$$\max_{S \neq \emptyset} G_f(S) = \max_{k \in [n]} J_k = J_{k^\star}. \tag{68}$$

Therefore, for every $k \in [|C|]$,

$$J_k^C = G_f(S_k^C) \leq \max_{S \neq \emptyset} G_f(S) = J_{k^\star} = J_{k^\star}^C. \tag{69}$$

So $k^\star$ is a maximizer of the pool scan. Finally, since we select the *smallest* maximizer in both scans and (67) shows $J_k^C = J_k$ on $k \in [L]$ with $L \geq k^\star$, we get $k_C^\star = k^\star$ and hence $S_C^\star = S^\star$. $\qquad \square$

**Example (what does `top_m=1200` mean?).** At a single decoding step, the model produces logits $\ell \in \mathbb{R}^n$ and hence a next-token distribution $p \in \Delta^{n-1}$ (after applying the selection temperature). The hyperparameter `top_m` restricts all Top-$W$ computations to a *candidate pool* $C \subseteq [n]$ consisting of the `top_m` most probable tokens under $p$:

$$C := \arg\operatorname{top} \operatorname{M}_{i \in [n]} p_i, \qquad |C| = \texttt{top\_m}. \tag{70}$$

Thus `top_m=1200` means that we keep only the 1200 indices with the largest $p_i$ values and discard the remaining $n - 1200$ tokens from consideration in the subset-update step.

Concretely, given the current set $S^{(t)}$, we compute the geometry-driven potential (e.g., $f_i^{(t)} = -\operatorname{dist}(i, S^{(t)})$) only for $i \in C$, form scores

$$\phi_i^{(t)} = f_i^{(t)} + \lambda \log p_i, \qquad i \in C, \tag{71}$$

sort *only within the pool* by decreasing $\phi_i^{(t)}$, scan prefixes over that pool, and return a crop $S^{(t+1)} \subseteq C$. Finally, logits outside the selected crop are masked to $-\infty$.

Theorem I.1 shows that this restriction can be *exact* for the fixed-$f$ subset step: if the full-vocabulary optimal prefix in $\phi$-order has size $k^\star$ and the pool contains the top-$L$ items in that $\phi$-order for some $L \geq k^\star$ (in particular, if it contains the top-$k^\star$ items), then the pool-restricted prefix scan returns the same optimizer as the full scan.

## J. Evaluation Protocol

For GSM8K/GPQA, we use exact-match accuracy under standard prompting. Comparisons are paired: each method is evaluated on the same question set and seeds. For AlpacaEval/MT-Bench and rubric scoring, we aggregate judge scores over repeated samplings; for the rubric, we score a fixed prompt set at each $T$ across multiple axes Diversity/Originality/Narrative Flow/Emotional Impact/Imagery, reporting mean±std. To reduce positional bias in judge-based settings, we randomize method order per item and aggregate across repeats.

## K. Time Complexity Analysis

Let $n := |\mathcal{V}|$ be the vocabulary size and $m$ be the candidate pool size (`top_m`). Let $d$ denote the embedding dimension, and let $I$ be the number of alternating updates (`alt_iters`; in our experiments we use a small constant, e.g., $I = 3$). All

bounds are stated per decoding step (one call of the logits processor).

**Theorem K.1** (Time complexity of Top-*W* (geom_mode = M)). *One Top-W decoding step runs in*

$$\mathcal{O}\big(n \log m + n + md + I\,(m^2 d + m \log m)\big).$$

*With I treated as a small constant (and d fixed for a given model), this simplifies to*

$$\mathcal{O}\big(n \log m + n + m^2 d\big),$$

*and in practice the runtime is typically dominated by the n-scale terms (pool selection, normalization) plus a modest m-scale geometric refinement.*

*Proof.* We decompose one decoding step into several standard operations.

(1) *Pool formation.* Selecting the top-$m$ logits from $n$ entries (partial sort / `topk`) costs $\mathcal{O}(n \log m)$.

(2) *Probability normalization.* Computing $\log Z = \log \sum_{i=1}^{n} e^{\ell_i}$ via `logsumexp` costs $\mathcal{O}(n)$, and converting the $m$ pooled logits to probabilities costs $\mathcal{O}(m)$.

(3) *Geometric preprocessing on the pool.* Gathering $m$ embeddings and whitening them costs $\mathcal{O}(md)$.

(4) *Alternating refinement (repeat I times).* Each iteration computes distances from all $m$ pooled items to the current selected set using a matrix product of shape $(m \times d)$ by $(d \times k)$ with $k \leq m$, which costs $\mathcal{O}(mdk) \subseteq \mathcal{O}(m^2 d)$. The subsequent prefix-selection step sorts $m$ scores, costing $\mathcal{O}(m \log m)$. Thus each iteration costs $\mathcal{O}(m^2 d + m \log m)$, and $I$ iterations cost $\mathcal{O}\big(I(m^2 d + m \log m)\big)$.

(5) *Masking.* Writing $-\infty$ to the logits vector touches $n$ entries, costing $\mathcal{O}(n)$.

Summing (1)–(5) yields $\mathcal{O}\big(n \log m + n + md + I(m^2 d + m \log m)\big)$. □

**Per-sequence cost.** For generating $L$ tokens, the total cost scales linearly in $L$:

$$\mathcal{O}\big(L\,(n \log m + n + md + I(m^2 d + m \log m))\big).$$

**Wall-clock decoding speed.** We also report empirical wall-clock decoding speed (milliseconds per generated token, ms/tok) for Top-*W*, Top-*H*, Top-*p*, and Min-*p* in Table 4. Across models and temperatures, Top-*W* is consistently close to the baselines, and is on average only slightly slower. Overall, the additional geometric refinement in Top-*W* leads to a modest overhead, so the practical decoding-time differences relative to Top-*H* and top-*p* (and similarly top-*k*) are small. Top-*W* is on average **about 5.4% slower** than Top-*H*, Top-*p*, and Min-*p*.

## L. Hyperparameter Tuning

We held out 50 random examples from AlpacaEval as a development set and performed the grid search only on that subset. After selecting the hyperparameters, we fixed them across all experiments. Specifically, we sweep $\lambda, \beta \in$ {0.0, 0.1, 0.25, 0.5, 0.75, 1.0, 1.25, 1.5, 1.75, 2.0, 2.25, 2.5, 2.75, 3.0, 3.5, 4.0, 4.5, 5.0}. On this coarse grid, the best-performing region is centered around $(\lambda, \beta) \approx (2.25, 2.75)$. We then run a narrower search around this region and select $(\lambda, \beta) = (2.2, 2.8)$ as our default setting for all main experiments.

## M. Results of LLM as a judge

### M.1. LLM-as-a-Judge evaluation prompts

Following previous research, (Nguyen et al., 2025) and (Baghaei Potraghloo et al., 2026) we design LLM as judge experiment as follows.

```
Judge Evaluation Prompt

You are an expert judge evaluating
AI-generated creative writing.  I am
testing the diversity and coherent
writing capabilities of three different
models.  I will paste three different
responses that were generated here.  Rate
responses based on the following metrics:
(1) Diversity:  Novelty and uniqueness of
ideas;
(2) Originality:  Innovative approach to
the prompt;
(3) Narrative Flow:  Coherence of the
text;
(4) Emotional Impact:  Ability to evoke
feelings;
(5) Imagery:  Vividness of descriptions.
Rate each metric from 1 to 10.  Also,
suggest the overall winner:  the response
that best maintains high coherence while
demonstrating high diversity.
```

```
Prompt 1

Write a story about an alien
civilization's first contact with Earth
from their perspective.
```

```
Prompt 2

Write a story about a world where time
suddenly starts moving backwards.
```

```
Prompt 3

Write a story about a mysterious door
that appears in an unexpected place.
```

| | Phi-3 (ms/tok) | | | | Qwen2.5-3B (ms/tok) | | | | LLaMA-3.1-8B (ms/tok) | | | |
|---|---|---|---|---|---|---|---|---|---|---|---|---|
| **Temp** | Top-*W* | Top-*H* | Min-*p* | Top-*p* | Top-*W* | Top-*H* | Min-*p* | Top-*p* | Top-*W* | Top-*H* | Min-*p* | Top-*p* |
| 1.0 | 24.3253 | 23.0895 | 23.0981 | 23.1436 | 25.2803 | 23.9940 | 24.0362 | 23.8861 | 26.8622 | 25.9304 | 25.9202 | 25.8799 |
| 2.0 | 24.6232 | 23.1370 | 23.0632 | 23.2854 | 25.3260 | 24.0831 | 24.0633 | 24.0998 | 27.6060 | 25.9268 | 25.9107 | 25.8495 |

*Table 4.* Decoding speed (milliseconds per generated token, ms/tok) for different sampling methods across temperatures.

## M.2. Results

Below, we go over per-model breakdown of Table 8. Judge-based evaluation metrics (scores from 1.0 to 10.0) across temperatures, prompts, and sampling methods are reported for (Table 5) LLaMA3.1-8B-Instruct, (Table 6) Phi-3-Mini and (Table 7) Qwen2.5-3B.

| Temp | Prompt | Sampling | M1 | M2 | M3 | M4 | M5 | Avg |
|------|--------|----------|------|------|------|------|------|------|
| 1.0 | Prompt 1 | Top-*p* | $7.00 \pm 0.89$ | $6.40 \pm 1.36$ | $7.60 \pm 0.80$ | $\mathbf{7.20 \pm 0.98}$ | $7.20 \pm 0.75$ | $7.08 \pm 0.74$ |
| | | Min-*p* | $7.20 \pm 1.17$ | $6.40 \pm 1.50$ | $\mathbf{7.80 \pm 0.40}$ | $7.20 \pm 0.98$ | $7.20 \pm 1.17$ | $7.16 \pm 0.98$ |
| | | Top-*H* | $\mathbf{7.50 \pm 1.12}$ | $\mathbf{7.00 \pm 1.00}$ | $7.75 \pm 0.43$ | $7.00 \pm 0.71$ | $\mathbf{7.25 \pm 0.83}$ | $\mathbf{7.30 \pm 0.75}$ |
| | | Top-*W* | $7.20 \pm 0.40$ | $6.40 \pm 0.49$ | $7.40 \pm 0.49$ | $6.60 \pm 0.49$ | $7.00 \pm 0.00$ | $6.92 \pm 0.16$ |
| | Prompt 2 | Top-*p* | $\mathbf{7.60 \pm 1.36}$ | $\mathbf{7.40 \pm 1.20}$ | $7.60 \pm 0.49$ | $\mathbf{7.40 \pm 0.80}$ | $\mathbf{7.80 \pm 0.75}$ | $\mathbf{7.56 \pm 0.82}$ |
| | | Min-*p* | $7.40 \pm 0.49$ | $6.60 \pm 0.80$ | $7.60 \pm 0.49$ | $6.60 \pm 0.49$ | $7.80 \pm 0.40$ | $7.20 \pm 0.13$ |
| | | Top-*H* | $6.80 \pm 0.40$ | $6.80 \pm 0.40$ | $7.60 \pm 0.49$ | $7.00 \pm 0.63$ | $7.40 \pm 0.49$ | $7.12 \pm 0.30$ |
| | | Top-*W* | $7.00 \pm 0.89$ | $7.20 \pm 0.98$ | $7.20 \pm 0.40$ | $7.00 \pm 0.89$ | $7.20 \pm 0.98$ | $7.12 \pm 0.64$ |
| | Prompt 3 | Top-*p* | $\mathbf{7.60 \pm 0.49}$ | $\mathbf{7.40 \pm 1.20}$ | $\mathbf{7.80 \pm 0.40}$ | $\mathbf{7.80 \pm 0.75}$ | $\mathbf{8.20 \pm 0.40}$ | $\mathbf{7.76 \pm 0.56}$ |
| | | Min-*p* | $6.50 \pm 0.87$ | $6.50 \pm 1.50$ | $7.50 \pm 0.50$ | $6.75 \pm 1.30$ | $7.25 \pm 0.43$ | $6.90 \pm 0.88$ |
| | | Top-*H* | $7.20 \pm 0.40$ | $6.80 \pm 0.75$ | $7.80 \pm 0.40$ | $7.40 \pm 0.49$ | $8.00 \pm 0.00$ | $7.44 \pm 0.23$ |
| | | Top-*W* | $6.75 \pm 1.30$ | $6.75 \pm 1.09$ | $7.50 \pm 0.50$ | $7.50 \pm 1.12$ | $7.50 \pm 0.87$ | $7.20 \pm 0.73$ |
| 1.5 | Prompt 1 | Top-*p* | $7.00 \pm 2.53$ | $6.60 \pm 2.73$ | $2.60 \pm 0.80$ | $3.60 \pm 1.36$ | $4.20 \pm 1.17$ | $4.80 \pm 1.66$ |
| | | Min-*p* | $\mathbf{7.60 \pm 0.49}$ | $\mathbf{7.80 \pm 0.75}$ | $\mathbf{8.60 \pm 0.49}$ | $\mathbf{7.60 \pm 0.49}$ | $\mathbf{8.00 \pm 0.63}$ | $\mathbf{7.92 \pm 0.41}$ |
| | | Top-*H* | $7.40 \pm 0.49$ | $7.40 \pm 1.02$ | $8.40 \pm 0.49$ | $7.20 \pm 1.17$ | $7.40 \pm 1.02$ | $7.56 \pm 0.74$ |
| | | Top-*W* | $6.40 \pm 0.49$ | $6.40 \pm 0.49$ | $8.20 \pm 0.40$ | $6.60 \pm 0.49$ | $6.40 \pm 0.49$ | $6.80 \pm 0.22$ |
| | Prompt 2 | Top-*p* | $3.80 \pm 0.98$ | $2.80 \pm 0.98$ | $1.60 \pm 0.80$ | $1.80 \pm 0.98$ | $2.40 \pm 1.02$ | $2.48 \pm 0.93$ |
| | | Min-*p* | $\mathbf{7.40 \pm 0.80}$ | $\mathbf{7.80 \pm 0.98}$ | $8.20 \pm 0.75$ | $7.00 \pm 2.10$ | $\mathbf{8.00 \pm 0.89}$ | $\mathbf{7.68 \pm 0.75}$ |
| | | Top-*H* | $7.20 \pm 0.40$ | $7.40 \pm 0.80$ | $\mathbf{8.40 \pm 0.49}$ | $\mathbf{7.60 \pm 0.49}$ | $7.80 \pm 0.40$ | $7.68 \pm 0.41$ |
| | | Top-*W* | $6.75 \pm 0.83$ | $6.75 \pm 0.83$ | $8.00 \pm 0.00$ | $7.25 \pm 1.09$ | $7.75 \pm 0.83$ | $7.30 \pm 0.67$ |
| | Prompt 3 | Top-*p* | $2.80 \pm 0.40$ | $2.00 \pm 0.00$ | $1.00 \pm 0.00$ | $1.20 \pm 0.40$ | $1.80 \pm 0.40$ | $1.76 \pm 0.20$ |
| | | Min-*p* | $6.60 \pm 0.80$ | $7.20 \pm 0.40$ | $\mathbf{8.20 \pm 0.40}$ | $7.60 \pm 0.49$ | $7.60 \pm 0.80$ | $7.44 \pm 0.51$ |
| | | Top-*H* | $\mathbf{7.40 \pm 0.49}$ | $\mathbf{7.40 \pm 0.80}$ | $7.80 \pm 0.75$ | $\mathbf{8.00 \pm 0.63}$ | $\mathbf{7.80 \pm 0.75}$ | $\mathbf{7.68 \pm 0.53}$ |
| | | Top-*W* | $6.25 \pm 0.43$ | $6.25 \pm 0.43$ | $8.00 \pm 0.00$ | $7.00 \pm 0.00$ | $7.50 \pm 0.50$ | $7.00 \pm 0.24$ |
| 2.0 | Prompt 1 | Top-*p* | $4.60 \pm 2.42$ | $3.80 \pm 2.79$ | $2.00 \pm 0.89$ | $2.20 \pm 1.17$ | $3.20 \pm 1.60$ | $3.16 \pm 1.67$ |
| | | Min-*p* | $\mathbf{7.40 \pm 0.80}$ | $\mathbf{7.80 \pm 1.47}$ | $6.20 \pm 1.94$ | $6.40 \pm 1.36$ | $6.80 \pm 1.17$ | $6.92 \pm 1.29$ |
| | | Top-*H* | $7.00 \pm 0.63$ | $7.20 \pm 0.75$ | $5.80 \pm 1.17$ | $5.20 \pm 1.17$ | $6.00 \pm 0.63$ | $6.24 \pm 0.66$ |
| | | Top-*W* | $6.80 \pm 0.75$ | $7.40 \pm 1.02$ | $\mathbf{8.00 \pm 0.63}$ | $\mathbf{7.00 \pm 0.89}$ | $\mathbf{7.20 \pm 0.40}$ | $\mathbf{7.28 \pm 0.41}$ |
| | Prompt 2 | Top-*p* | $4.20 \pm 2.71$ | $3.20 \pm 2.86$ | $1.40 \pm 0.49$ | $1.80 \pm 0.98$ | $2.80 \pm 2.40$ | $2.68 \pm 1.87$ |
| | | Min-*p* | $7.00 \pm 0.63$ | $6.40 \pm 0.80$ | $5.40 \pm 1.62$ | $5.60 \pm 1.02$ | $6.20 \pm 0.98$ | $6.12 \pm 0.93$ |
| | | Top-*H* | $5.80 \pm 0.75$ | $5.20 \pm 0.75$ | $4.00 \pm 0.63$ | $4.80 \pm 0.98$ | $4.60 \pm 0.80$ | $4.88 \pm 0.72$ |
| | | Top-*W* | $\mathbf{7.40 \pm 0.80}$ | $\mathbf{8.00 \pm 0.63}$ | $\mathbf{8.60 \pm 0.49}$ | $\mathbf{7.80 \pm 0.40}$ | $\mathbf{7.60 \pm 0.80}$ | $\mathbf{7.88 \pm 0.56}$ |
| | Prompt 3 | Top-*p* | $2.00 \pm 0.00$ | $1.60 \pm 0.49$ | $1.00 \pm 0.00$ | $1.00 \pm 0.00$ | $1.00 \pm 0.00$ | $1.32 \pm 0.10$ |
| | | Min-*p* | $5.60 \pm 0.49$ | $5.00 \pm 0.63$ | $3.80 \pm 1.60$ | $4.00 \pm 0.63$ | $5.00 \pm 0.63$ | $4.68 \pm 0.64$ |
| | | Top-*H* | $5.20 \pm 1.17$ | $5.20 \pm 1.72$ | $3.00 \pm 0.89$ | $3.80 \pm 0.98$ | $4.60 \pm 1.20$ | $4.36 \pm 1.09$ |
| | | Top-*W* | $\mathbf{7.00 \pm 0.63}$ | $\mathbf{7.60 \pm 0.49}$ | $\mathbf{8.60 \pm 0.49}$ | $\mathbf{7.60 \pm 0.49}$ | $\mathbf{8.00 \pm 0.00}$ | $\mathbf{7.76 \pm 0.32}$ |

*Table 5.* LLM-as-a-Judge creative-writing scores for Llama-3.1-8B-Instruct. Each entry is mean $\pm$ std over 5 repeats; higher is better. For each (temperature, prompt), the maximum value in each metric column is bolded (ties broken by method order: Top-*p*, Min-*p*, Top-*H*, Top-*W*).

| Temp | Prompt | Sampling | M1 | M2 | M3 | M4 | M5 | Avg |
|---|---|---|---|---|---|---|---|---|
| 1.0 | Prompt 1 | Top-*p* | **7.40 ± 0.49** | **7.00 ± 0.89** | 7.20 ± 0.98 | **7.60 ± 0.49** | **8.40 ± 0.49** | **7.52 ± 0.35** |
| | | Min-*p* | 6.60 ± 0.80 | 6.00 ± 0.89 | **8.00 ± 0.00** | 6.80 ± 0.75 | 7.60 ± 0.80 | 7.00 ± 0.61 |
| | | Top-*H* | 6.00 ± 0.00 | 5.80 ± 0.75 | 7.80 ± 0.40 | 6.60 ± 0.80 | 7.00 ± 0.00 | 6.64 ± 0.23 |
| | | Top-*W* | 7.00 ± 0.63 | 7.00 ± 1.10 | 7.80 ± 0.40 | 7.20 ± 0.98 | 7.60 ± 0.80 | 7.32 ± 0.59 |
| | Prompt 2 | Top-*p* | **8.00 ± 1.26** | **7.60 ± 1.50** | 7.80 ± 0.75 | **8.00 ± 1.10** | **8.00 ± 1.26** | **7.88 ± 1.06** |
| | | Min-*p* | 7.00 ± 1.10 | 6.80 ± 1.17 | 7.60 ± 1.02 | 7.00 ± 1.10 | 7.00 ± 0.89 | 7.08 ± 0.78 |
| | | Top-*H* | 7.40 ± 0.80 | 6.80 ± 0.75 | **8.20 ± 0.75** | 7.60 ± 0.49 | 7.40 ± 0.49 | 7.48 ± 0.59 |
| | | Top-*W* | 7.00 ± 1.41 | 6.40 ± 1.02 | 7.60 ± 0.49 | 6.60 ± 0.49 | 6.60 ± 1.02 | 6.84 ± 0.79 |
| | Prompt 3 | Top-*p* | 7.00 ± 0.63 | **6.60 ± 1.02** | 7.60 ± 0.49 | 6.80 ± 0.40 | 7.80 ± 0.75 | 7.16 ± 0.57 |
| | | Min-*p* | 7.00 ± 0.63 | 6.20 ± 0.98 | 7.20 ± 0.40 | 7.00 ± 0.63 | 7.80 ± 0.75 | 7.04 ± 0.65 |
| | | Top-*H* | 7.00 ± 0.63 | 6.20 ± 0.98 | **7.80 ± 0.40** | 7.00 ± 0.63 | 7.60 ± 0.49 | 7.12 ± 0.45 |
| | | Top-*W* | **7.25 ± 0.83** | 6.25 ± 0.83 | 7.75 ± 0.43 | **7.25 ± 0.83** | **8.00 ± 0.71** | **7.30 ± 0.70** |
| 1.5 | Prompt 1 | Top-*p* | 4.40 ± 1.02 | 3.40 ± 1.02 | 2.00 ± 0.63 | 2.40 ± 1.02 | 3.00 ± 0.63 | 3.04 ± 0.85 |
| | | Min-*p* | **8.00 ± 1.10** | **7.80 ± 1.47** | **8.40 ± 0.49** | **8.60 ± 0.80** | **8.20 ± 0.75** | **8.20 ± 0.82** |
| | | Top-*H* | 7.40 ± 0.49 | 7.00 ± 0.63 | 8.20 ± 0.40 | 8.20 ± 0.75 | 7.60 ± 0.80 | 7.68 ± 0.53 |
| | | Top-*W* | 6.60 ± 0.80 | 6.00 ± 0.63 | 8.00 ± 0.63 | 7.40 ± 0.49 | 7.20 ± 0.75 | 7.04 ± 0.56 |
| | Prompt 2 | Top-*p* | 5.80 ± 1.47 | 5.20 ± 1.94 | 2.40 ± 0.49 | 3.20 ± 0.75 | 4.40 ± 1.36 | 4.20 ± 1.17 |
| | | Min-*p* | **7.00 ± 0.63** | **7.40 ± 1.02** | **8.40 ± 0.49** | **8.00 ± 0.63** | **7.80 ± 0.75** | **7.72 ± 0.55** |
| | | Top-*H* | 7.00 ± 0.89 | 7.20 ± 0.75 | 7.80 ± 0.40 | 7.60 ± 0.80 | 7.40 ± 0.80 | 7.40 ± 0.55 |
| | | Top-*W* | 6.80 ± 0.40 | 7.20 ± 0.75 | 8.40 ± 0.49 | 7.60 ± 0.49 | 7.20 ± 0.40 | 7.44 ± 0.32 |
| | Prompt 3 | Top-*p* | 5.40 ± 2.58 | 5.00 ± 3.29 | 2.20 ± 1.17 | 2.80 ± 1.47 | 3.60 ± 1.62 | 3.80 ± 1.99 |
| | | Min-*p* | 6.80 ± 0.40 | 6.20 ± 0.98 | 8.00 ± 0.89 | 7.20 ± 1.17 | 7.40 ± 0.80 | 7.12 ± 0.74 |
| | | Top-*H* | **7.20 ± 0.75** | 6.60 ± 1.02 | 7.80 ± 0.98 | **7.40 ± 0.49** | 7.80 ± 0.75 | 7.36 ± 0.61 |
| | | Top-*W* | 7.00 ± 0.89 | **6.60 ± 1.36** | **8.20 ± 0.75** | 7.20 ± 0.75 | **8.00 ± 0.89** | **7.40 ± 0.90** |
| 2.0 | Prompt 1 | Top-*p* | 3.20 ± 1.47 | 2.40 ± 1.36 | 1.40 ± 0.80 | 1.60 ± 1.20 | 2.00 ± 1.55 | 2.12 ± 1.26 |
| | | Min-*p* | 7.20 ± 0.40 | 7.00 ± 0.63 | 7.40 ± 0.80 | 6.20 ± 0.75 | 6.80 ± 0.40 | 6.92 ± 0.37 |
| | | Top-*H* | **8.20 ± 0.75** | **8.20 ± 1.17** | 6.00 ± 0.63 | 7.20 ± 0.75 | 8.00 ± 1.10 | 7.52 ± 0.77 |
| | | Top-*W* | 7.75 ± 0.43 | 7.75 ± 1.09 | **8.75 ± 0.43** | **7.75 ± 0.43** | **8.50 ± 0.50** | **8.10 ± 0.44** |
| | Prompt 2 | Top-*p* | 2.40 ± 0.49 | 1.60 ± 0.49 | 1.00 ± 0.00 | 1.00 ± 0.00 | 1.20 ± 0.40 | 1.44 ± 0.23 |
| | | Min-*p* | **8.25 ± 0.43** | **8.50 ± 0.50** | 8.25 ± 0.43 | **8.25 ± 0.83** | **8.00 ± 0.71** | **8.25 ± 0.30** |
| | | Top-*H* | 7.00 ± 0.89 | 7.20 ± 1.33 | 5.00 ± 1.10 | 6.20 ± 1.17 | 6.60 ± 1.20 | 6.40 ± 1.02 |
| | | Top-*W* | 7.60 ± 0.80 | 8.20 ± 0.75 | **8.60 ± 0.49** | 8.00 ± 0.63 | 7.60 ± 0.80 | 8.00 ± 0.49 |
| | Prompt 3 | Top-*p* | 2.20 ± 0.40 | 1.40 ± 0.49 | 1.00 ± 0.00 | 1.00 ± 0.00 | 1.20 ± 0.40 | 1.36 ± 0.23 |
| | | Min-*p* | **7.60 ± 0.49** | **7.60 ± 0.49** | 7.60 ± 1.50 | 7.20 ± 0.75 | 7.80 ± 0.75 | 7.56 ± 0.56 |
| | | Top-*H* | 6.20 ± 0.75 | 5.80 ± 1.33 | 5.00 ± 1.79 | 5.60 ± 1.62 | 6.20 ± 1.72 | 5.76 ± 1.41 |
| | | Top-*W* | 7.20 ± 0.75 | 7.40 ± 1.36 | **8.40 ± 0.49** | 7.40 ± 0.49 | 8.20 ± 0.75 | 7.72 ± 0.69 |

*Table 6.* LLM-as-a-Judge creative-writing scores for Phi-3-mini-4k-instruct. Each entry is mean ± std over 5 repeats; higher is better. For each (temperature, prompt), the maximum value in each metric column is bolded (ties broken by method order: Top-*p*, Min-*p*, Top-*H*, Top-*W*).

| Temp | Prompt | Sampling | M1 | M2 | M3 | M4 | M5 | Avg |
|---|---|---|---|---|---|---|---|---|
| 1.0 | Prompt 1 | Top-$p$ | 5.80 ± 1.94 | 5.40 ± 1.85 | 6.20 ± 2.14 | 5.80 ± 2.04 | 6.40 ± 2.58 | 5.92 ± 2.02 |
| | | Min-$p$ | 6.40 ± 0.80 | 5.60 ± 1.20 | 7.00 ± 1.10 | 6.00 ± 1.10 | 6.60 ± 0.80 | 6.32 ± 0.78 |
| | | Top-$H$ | 5.80 ± 1.17 | 4.80 ± 1.17 | 6.40 ± 2.24 | 5.40 ± 1.36 | 6.00 ± 1.79 | 5.68 ± 1.49 |
| | | Top-$W$ | **7.00 ± 0.63** | **6.40 ± 1.02** | **8.00 ± 0.63** | **7.20 ± 0.75** | **7.60 ± 0.49** | **7.24 ± 0.59** |
| | Prompt 2 | Top-$p$ | **7.00 ± 1.10** | **7.40 ± 0.80** | 7.20 ± 0.75 | **7.20 ± 0.98** | **7.00 ± 0.63** | **7.16 ± 0.71** |
| | | Min-$p$ | 6.40 ± 1.20 | 6.40 ± 1.20 | 7.00 ± 0.89 | 6.80 ± 1.17 | 6.60 ± 1.36 | 6.64 ± 1.11 |
| | | Top-$H$ | 6.20 ± 0.40 | 5.80 ± 0.75 | **7.40 ± 0.49** | 6.80 ± 0.75 | 6.40 ± 0.49 | 6.52 ± 0.45 |
| | | Top-$W$ | 6.40 ± 1.02 | 6.20 ± 0.98 | 7.00 ± 0.63 | 6.60 ± 0.80 | 6.20 ± 0.75 | 6.48 ± 0.75 |
| | Prompt 3 | Top-$p$ | **7.00 ± 0.89** | 7.20 ± 1.17 | 7.00 ± 0.89 | 6.60 ± 1.02 | **7.80 ± 0.40** | 7.12 ± 0.68 |
| | | Min-$p$ | 6.60 ± 1.02 | 6.40 ± 1.36 | 7.40 ± 0.49 | **7.00 ± 1.10** | 7.60 ± 1.02 | 7.00 ± 0.87 |
| | | Top-$H$ | 7.00 ± 0.71 | 6.75 ± 1.30 | **7.75 ± 0.83** | 6.75 ± 0.83 | 7.50 ± 1.12 | **7.15 ± 0.85** |
| | | Top-$W$ | 6.00 ± 0.89 | 6.20 ± 0.98 | 7.60 ± 0.49 | 6.40 ± 0.80 | 7.20 ± 0.75 | 6.68 ± 0.68 |
| 1.5 | Prompt 1 | Top-$p$ | 1.75 ± 0.43 | 1.50 ± 0.50 | 1.00 ± 0.00 | 1.00 ± 0.00 | 1.00 ± 0.00 | 1.25 ± 0.17 |
| | | Min-$p$ | 5.75 ± 0.83 | 5.75 ± 0.83 | 6.50 ± 1.50 | 5.50 ± 1.50 | 5.75 ± 1.64 | 5.85 ± 1.18 |
| | | Top-$H$ | 5.50 ± 1.50 | 5.50 ± 1.12 | 5.25 ± 2.17 | 4.50 ± 1.80 | 5.50 ± 1.80 | 5.25 ± 1.64 |
| | | Top-$W$ | **7.00 ± 1.22** | **7.25 ± 1.30** | **8.25 ± 0.83** | **7.50 ± 0.87** | **8.25 ± 1.30** | **7.65 ± 1.07** |
| | Prompt 2 | Top-$p$ | 1.60 ± 0.49 | 1.20 ± 0.40 | 1.00 ± 0.00 | 1.00 ± 0.00 | 1.00 ± 0.00 | 1.16 ± 0.15 |
| | | Min-$p$ | 6.60 ± 0.80 | 6.00 ± 0.89 | 7.40 ± 1.02 | 7.00 ± 0.89 | 6.60 ± 1.02 | 6.72 ± 0.84 |
| | | Top-$H$ | **7.20 ± 0.98** | **6.80 ± 0.75** | 6.80 ± 1.17 | **7.20 ± 0.98** | **7.20 ± 0.98** | **7.04 ± 0.90** |
| | | Top-$W$ | 6.40 ± 1.85 | 6.20 ± 1.60 | **7.60 ± 1.36** | 6.40 ± 1.36 | 6.40 ± 1.85 | 6.60 ± 1.56 |
| | Prompt 3 | Top-$p$ | 3.00 ± 2.53 | 2.20 ± 1.47 | 1.00 ± 0.00 | 1.20 ± 0.40 | 1.40 ± 0.80 | 1.76 ± 1.03 |
| | | Min-$p$ | **7.00 ± 0.63** | **7.40 ± 0.80** | **8.00 ± 0.63** | 7.20 ± 0.75 | **7.60 ± 0.49** | **7.44 ± 0.54** |
| | | Top-$H$ | 7.00 ± 0.89 | 6.80 ± 1.47 | 7.60 ± 0.80 | **7.40 ± 0.80** | 7.60 ± 0.80 | 7.28 ± 0.84 |
| | | Top-$W$ | 6.80 ± 0.75 | 6.20 ± 0.98 | 7.80 ± 0.40 | 6.60 ± 0.49 | 7.40 ± 0.80 | 6.96 ± 0.62 |
| 2.0 | Prompt 1 | Top-$p$ | 2.20 ± 1.94 | 2.00 ± 1.55 | 1.20 ± 0.40 | 1.20 ± 0.40 | 1.40 ± 0.80 | 1.60 ± 1.01 |
| | | Min-$p$ | 5.80 ± 2.14 | 5.60 ± 2.42 | 5.40 ± 1.85 | 4.80 ± 2.32 | 5.60 ± 2.65 | 5.44 ± 2.21 |
| | | Top-$H$ | 5.60 ± 1.62 | 5.20 ± 1.33 | 3.40 ± 0.80 | 3.20 ± 0.75 | 4.40 ± 0.80 | 4.36 ± 0.96 |
| | | Top-$W$ | **7.80 ± 0.75** | **7.80 ± 0.98** | **8.20 ± 0.75** | **7.60 ± 0.49** | **7.60 ± 0.49** | **7.80 ± 0.40** |
| | Prompt 2 | Top-$p$ | 2.00 ± 1.55 | 2.00 ± 2.00 | 1.20 ± 0.40 | 1.40 ± 0.80 | 1.60 ± 1.20 | 1.64 ± 1.18 |
| | | Min-$p$ | **6.80 ± 0.98** | **7.00 ± 1.26** | 6.20 ± 1.33 | 6.00 ± 1.26 | 6.20 ± 0.98 | 6.44 ± 1.00 |
| | | Top-$H$ | 5.00 ± 2.61 | 5.00 ± 3.16 | 2.60 ± 1.36 | 3.60 ± 1.85 | 4.80 ± 2.99 | 4.20 ± 2.36 |
| | | Top-$W$ | 6.40 ± 1.62 | 6.40 ± 2.15 | **7.40 ± 1.74** | **6.40 ± 1.85** | **7.40 ± 1.02** | **6.80 ± 1.56** |
| | Prompt 3 | Top-$p$ | 1.60 ± 0.49 | 1.40 ± 0.49 | 1.00 ± 0.00 | 1.00 ± 0.00 | 1.00 ± 0.00 | 1.20 ± 0.18 |
| | | Min-$p$ | **6.40 ± 0.80** | 6.00 ± 1.26 | 6.00 ± 0.63 | 5.80 ± 0.75 | 6.60 ± 0.80 | 6.16 ± 0.65 |
| | | Top-$H$ | 5.20 ± 0.75 | 4.60 ± 0.49 | 2.80 ± 0.40 | 3.40 ± 0.49 | 4.20 ± 0.75 | 4.04 ± 0.48 |
| | | Top-$W$ | 6.40 ± 0.49 | **6.80 ± 0.98** | **8.00 ± 0.00** | **6.80 ± 0.40** | **7.60 ± 0.80** | **7.12 ± 0.48** |

*Table 7.* LLM-as-a-Judge creative-writing scores for Qwen2.5-3B. Each entry is mean ± std over 5 repeats; higher is better. For each (temperature, prompt), the maximum value in each metric column is bolded (ties broken by method order: Top-$p$, Min-$p$, Top-$H$, Top-$W$).

| | | Qwen | | | | Llama | | | | Phi | | | |
|---|---|---|---|---|---|---|---|---|---|---|---|---|---|
| $T$ | Prompt | Min-$p$ | Top-$p$ | Top-$H$ | Top-$W$ | Min-$p$ | Top-$p$ | Top-$H$ | Top-$W$ | Min-$p$ | Top-$p$ | Top-$H$ | Top-$W$ |
| 1.0 | Prompt 1 | 6.32 ± 0.78 | 5.92 ± 2.02 | 5.68 ± 1.49 | **7.24 ± 0.59** | 7.16 ± 0.98 | 7.08 ± 0.74 | **7.30 ± 0.75** | 6.92 ± 0.16 | 7.00 ± 0.61 | **7.52 ± 0.35** | 6.64 ± 0.23 | 7.32 ± 0.59 |
| 1.0 | Prompt 2 | 6.64 ± 1.11 | **7.16 ± 0.71** | 6.52 ± 0.45 | 6.48 ± 0.75 | 7.20 ± 0.13 | **7.56 ± 0.82** | 7.12 ± 0.30 | 7.12 ± 0.64 | 7.08 ± 0.78 | **7.88 ± 1.06** | 7.48 ± 0.59 | 6.84 ± 0.79 |
| 1.0 | Prompt 3 | 7.00 ± 0.87 | 7.12 ± 0.68 | **7.15 ± 0.85** | 6.68 ± 0.68 | 6.90 ± 0.88 | **7.76 ± 0.56** | 7.44 ± 0.23 | 7.20 ± 0.73 | 7.04 ± 0.65 | 7.16 ± 0.57 | 7.12 ± 0.45 | **7.30 ± 0.70** |
| 1.5 | Prompt 1 | 5.85 ± 1.18 | 1.25 ± 0.17 | 5.25 ± 1.64 | **7.65 ± 1.07** | **7.92 ± 0.41** | 4.80 ± 1.66 | 7.56 ± 0.74 | 6.80 ± 0.22 | **8.20 ± 0.82** | 3.04 ± 0.85 | 7.68 ± 0.53 | 7.04 ± 0.56 |
| 1.5 | Prompt 2 | 6.72 ± 0.84 | 1.16 ± 0.15 | **7.04 ± 0.90** | 6.60 ± 1.56 | 7.68 ± 0.75 | 2.48 ± 0.93 | **7.68 ± 0.41** | 7.30 ± 0.67 | **7.72 ± 0.55** | 4.20 ± 1.17 | 7.40 ± 0.55 | 7.44 ± 0.32 |
| 1.5 | Prompt 3 | **7.44 ± 0.54** | 1.76 ± 1.03 | 7.28 ± 0.84 | 6.96 ± 0.62 | 7.44 ± 0.51 | 1.76 ± 0.20 | **7.68 ± 0.53** | 7.00 ± 0.24 | 7.12 ± 0.74 | 3.80 ± 1.99 | 7.36 ± 0.61 | **7.40 ± 0.90** |
| 2.0 | Prompt 1 | 5.44 ± 2.21 | 1.60 ± 1.01 | 4.36 ± 0.96 | **7.80 ± 0.40** | 6.92 ± 1.29 | 3.16 ± 1.67 | 6.24 ± 0.66 | **7.28 ± 0.41** | 6.92 ± 0.37 | 2.12 ± 1.26 | 7.52 ± 0.77 | **8.10 ± 0.44** |
| 2.0 | Prompt 2 | 6.44 ± 1.00 | 1.64 ± 1.18 | 4.20 ± 2.36 | **6.80 ± 1.56** | 6.12 ± 0.93 | 2.68 ± 1.87 | 4.88 ± 0.72 | **7.88 ± 0.56** | **8.25 ± 0.30** | 1.44 ± 0.23 | 6.40 ± 1.02 | 8.00 ± 0.49 |
| 2.0 | Prompt 3 | 6.16 ± 0.65 | 1.20 ± 0.18 | 4.04 ± 0.48 | **7.12 ± 0.48** | 4.68 ± 0.64 | 1.32 ± 0.10 | 4.36 ± 1.09 | **7.76 ± 0.32** | 7.56 ± 0.56 | 1.36 ± 0.23 | 5.76 ± 1.41 | **7.72 ± 0.69** |

*Table 8.* Average judge score (mean ± std over repeats) for each temperature $T$ and prompt, comparing four sampling methods on three base models. For each ($T$, prompt) and model, the best (highest mean) method is bolded; ties are bolded for all maxima. For details on each of the, Diversity, Originality, Narrative Flow, Emotional Impact check the Appendix. Across all 27 triplets, Min-$p$, top-$p$, and Top-$H$ win 6, 5, and 5 cases respectively, while Top-$W$ wins 12. (This experiment is done under $\beta(T) = 1.5 + 1.5T$ while keeping the rest of parameters the same. In this experiment, $\beta$ is higher than the default of this paper which is $\beta = 2.8$ and we can see that higher $\beta$ can actually help depending on dataset and type of experiment such as when we value Diversity, Originality, Narrative Flow, Emotional Impact check, all together.)

