# OpenReview forum: "Geometry-Aware Decoding with Wasserstein-Regularized Truncation and Mass Penalties for Large Language Models"
_ICML.cc/2026/Conference — ICML 2026 spotlight_

### Official Review · Reviewer_SUUE · 2026-02-16

**Soundness:** 3
**Presentation:** 3
**Significance:** 3
**Originality:** 3
**Overall Recommendation:** 4
**Confidence:** 4

**Summary:**

This paper proposes Top-W, a geometry-aware decoding method for large
language models that goes beyond probability-based truncation methods
such as Top-k and Top-p.

Instead of selecting tokens solely based on their probabilities, Top-W
jointly optimizes:

-The Wasserstein distance between the original next-token distribution and the truncated distribution (capturing semantic geometry),
-The entropy of the truncated distribution (controlling diversity),
-The retained probability mass (preventing over-truncation).

The authors prove that the optimal truncated set has a simple
structure (either a prefix or a singleton under certain parameter
regimes), enabling efficient implementation.

Empirically, Top-W improves stability in reasoning tasks at high temperatures and enhances the balance between creativity and coherence in open-ended generation.

**Compliance With Llm Reviewing Policy:**

Affirmed.

**Final Justification:**

The rebuttal explained my concerns fully, therefore, I'll raise my final scores.

**Key Questions For Authors:**

-In typical deployments, decoding temperature is often set below 1.0;
what measurable gains does the proposed method provide in this more
realistic low-temperature regime?

-What is the essential distinction between Top-W and Top-H, and are
 the improvements truly due to geometric regularization beyond entropy
 control?

**Limitations:**

The paper does not provide a dedicated discussion of methodological limitations. While it includes an Impact Statement addressing potential societal risks and briefly analyzes computational overhead, it offers little reflection on technical constraints such as the reliance on embedding geometry, the absence of evaluation in lower-temperature regimes, or the extent to which improvements are distinct from entropy-based methods like Top-H. A more explicit limitations discussion would strengthen the paper.

The empirical evaluation primarily focuses on relatively high
temperature settings (T ≥ 1.0), whereas many real-world deployments
operate at lower temperatures (T < 1.0). As a result, it remains
unclear whether the proposed method delivers meaningful advantages in
the more commonly used low-temperature regime.

**Strengths And Weaknesses:**

The strength of this paper is in its a principled geometry-aware
decoding objective that unifies Wasserstein regularization, entropy
control, and mass retention, supported by strong theoretical
guarantees (e.g., prefix optimality) enabling efficient
implementation. It shows consistent empirical gains, particularly
improving stability at higher temperatures while preserving
creativity, making the contribution both theoretically grounded and
practically relevant.

The empirical gains appear most pronounced at higher temperatures,
where standard probability-based methods tend to degrade, raising the
question of whether the improvements stem primarily from stabilizing
high-temperature noise rather than offering a fundamentally different
decoding principle. At lower temperatures—where many practical
deployments operate—the benefits seem less clear or potentially
marginal. This temperature-dependent effectiveness suggests that the
method may function more as a robustness mechanism for stochastic
regimes than as a universally superior decoding strategy.

---

> ### Author Rebuttal · Authors · 2026-03-31
>
> We sincerely thank the reviewer for the thoughtful feedback and for recognizing that the paper provides a principled geometry-aware decoding objective, strong theoretical structure, and practically relevant empirical gains. We have made every effort to address the concerns. If the reviewer takes these points into consideration, we would be grateful if you could adjust your score in a positive way.
>
> **Gains in low-temperatures:**
>
> We thank the reviewer for this important point. We agree that the original submission emphasized on higher temperatures and did not sufficiently evaluate more conventional settings, i.e., Temperature<1.
>
> Our initial focus on T $\in$ {1.0, 1.5, 2.0} was motivated by prior work, especially Top-H, and by the fact that differences between truncation-based logits processors tend to shrink as temperature decreases. But per your suggestion we added the experiment for T $\in$ {0.5, 0.7} over GSM8K, GPQA and AlpacaEval, (we skipped MT-Bench as we couldn’t finish it in time) for all three models and all four decoding methods.
>
> GSM8K:
>
> | T = 0.5 | Qwen | LLaMA | Phi |
> |---|---|---|---|
> | Min-p | **75.8** | 75.3 | 84.5 |
> | Top-p | 74.8 | 75.6 | 85.1 |
> | Top-H | 74.9 | 77.1 | 85.4 |
> | Top-W | 75.4 | **77.2** | **85.9** |
>
> | T = 0.7  | Qwen | LLaMA | Phi |
> |---|---|---|---|
> | Min-p | 74.5 | 73.1 | 83.6 |
> | Top-p | 75.7 | 73.1 | 84.7 |
> | Top-H | 74.4 | **77.6** | 84.6 |
> | Top-W | **75.8** | 77.1 | **85.4** |
>
>
> GPQA:
>
> | T = 0.5 | Qwen | LLaMA | Phi |
> |---|---|---|---|
> | Min-p | 29.2 | **31.9** | 30.9 |
> | Top-p | **30.5** | 29.2 | 31.8 |
> | Top-H | 27.6 | 30.3 | **32.1** |
> | Top-W | **30.5** | 30.8 | **32.1** |
>
> | T = 0.7  | Qwen | LLaMA | Phi |
> |---|---|---|---|
> | Min-p | 27.0 | 30.1 | 28.5 |
> | Top-p | 27.4 | 29.2 | 29.9 |
> | Top-H | 27.6 | **30.5** | 32.5 |
> | Top-W | **31.4** | 29.2 | **33.0** |
>
>
> AlpacaEval:
>
> | T = 0.5 | Qwen | LLaMA | Phi |
> |---|---|---|---|
> | Min-p | 10.7 | 26.9 | 21.7 |
> | Top-p | **11.1** | 26.9 | 21.2 |
> | Top-H | 10.2 | 26.4 | 23.1 |
> | Top-W | 10.0 | **27.0** | **24.8** |
>
> | T = 0.7 | Qwen | LLaMA | Phi |
> |---|---|---|---|
> | Min-p | 9.1 | 25.8 | 22.6 |
> | Top-p | 7.2 | 26.9 | 21.2 |
> | Top-H | 8.1 | **28.6** | 21.8 |
> | Top-W | **9.7** | 26.7 | **23.1** |
>
>
> These new results show a more nuanced picture. At T$\in${0.5,0.7}, Top-W no longer dominates nearly all (T,model) settings as it did in the original higher-temperature results, which is expected because truncation-based methods become more similar at low temperature. Encouragingly, Top-W remains strongest overall across the completed low-temperature reruns: Min-p, Top-p, Top-H and Top-W win in 1,0,1,4 cases in GSM8K, 1,1,2,4 cases in GPQA, and 0,1,1,4 cases in AlpacaEval, respectively. We will include these results in the revised version of the paper.
>
>
> **Distinction between Top-W and Top-H and impact of geometry:**
>
> We thank the reviewer for this important point. The key difference is that Top-H is purely probability/entropy based, while Top-W additionally penalizes geometric distortion through a transport term defined over token embeddings. As a result, two candidate crops with similar retained mass and entropy can still be scored differently by Top-W if one perturbs the original distribution more severely in embedding space. In that sense, Top-W extends entropy-based truncation with an explicit notion of token-space geometry.
>
> This is also reflected in the theory (Section 4.3. in the paper). Under the uniform (0-1) metric, where transport ignores geometry, Top-W reduces to a probability-only criterion and recovers standard truncation rules as special cases, such as Top-$k$ and the Lagrangian form underlying Top-H. Thus, geometry is exactly what makes Top-W different.
>
> To make the role of geometry more explicit, we added a controlled ablation the same as the AlpacaEval experiment in the paper for phi-3-mini-4k-instruct at T $\in$ {0.5,1.0}. We compared Top-W under three ground metrics: a uniform / constant metric (geometry-free control), an L1 metric (simple geometry baseline), and the paper’s default diagonal Mahalanobis metric.
>
> Please **refer to our response to Reviewer 2TUp** for detailed results in the geometry-comparison table.
>
> The results show that the choice of geometry matters: the geometry-free control is weaker than both the L1 metric and the diagonal Mahalanobis metric, while the diagonal Mahalanobis metric is slightly better than the L1 metric. Overall, these results suggest that the gain comes from introducing meaningful token-space geometry, with the diagonal Mahalanobis metric giving the strongest performance. This supports our claim that OT helps by making truncation sensitive to embedding-space proximity, not just probability mass and entropy. We will add these results and discussions in the revised version of the paper.
>
>
> Thank you for taking the time to read and review our paper. We are grateful for your feedback, which substantially improved both the evaluation and presentation of the paper.

---

> > ### Author Rebuttal · Reviewer_SUUE · 2026-04-03
> >
> > The authors addressed my concerns fully. Will raise the score.

---

### Official Review · Reviewer_XuZF · 2026-02-21

**Soundness:** 3
**Presentation:** 2
**Significance:** 2
**Originality:** 3
**Overall Recommendation:** 4
**Confidence:** 2

**Summary:**

This work proposes optimal transport inspired probability truncation for language model sampling. The optimization objective involves 3 terms - Wasserstein distance penalizing distributional distortion, entropy penalty and probability mass reward. The Top-W subset is computed via alternating update rule. The introduced approach is validated on several language models - Llama-3.1/Qwen-2.5/Phi-3 and compared to existing sampling (Top-p, Min-p, Top-H) strategies at various temperatures.

**Compliance With Llm Reviewing Policy:**

Affirmed.

**Final Justification:**

After taking into account the rebuttal and author responses I decided to raise the score to Weak Accept.
The proposed sampling technique looks sound and theoretically grounded.

**Key Questions For Authors:**

- Have you measured the diversity of the responses generated with Top-W sampling and the baselines? I would recommend measuring Vendi Score [1] or a similar measure. Could it be the case, that Top-W sampling is effectively equivalent to some of the baselines with lower temperature?
- It would be good to evaluate the efficacy of the method on some reasoning task (AIME, AMC) with a reasoning model (DeepSeek-R1 distilled or from Qwen-3 model family).
---
References

[1] Friedman, Dan, and Adji Bousso Dieng. "The vendi score: A diversity evaluation metric for machine learning." arXiv preprint arXiv:2210.02410 (2022).

**Limitations:**

-

**Strengths And Weaknesses:**

Strengths
- The introduced method approach appears to be theoretically grounded with clear interpretation of the constituent terms in the objective.
- Top-W decoding is more stable than alternatives with the increase of temperature and consistently achieves higher accuracy.

Weaknesses
- The temperature grid is limited to relatively high temperatures (1.0-2.0) [1, 2]. At the same time, recommended temperatures for most models are below 1.0 - typically in the range 0.6-0.7. Does the Top-W sampling still produce best results at lower temperature?
---
References

[1] https://huggingface.co/Qwen/Qwen3-8B

[2] https://huggingface.co/unsloth/Meta-Llama-3.1-8B-Instruct/blob/main/generation_config.json

---

> ### Author Rebuttal · Authors · 2026-03-31
>
> We sincerely thank the reviewer for the thoughtful review and for recognizing both the theoretical grounding of the method and its stability advantages as temperature increases. We have made every effort to address the concerns and offer detailed explanations. If the reviewer takes these points into consideration and finds more value in our study, we would be grateful if you could adjust your score in a positive way.
>
> **Limited temperature grid:**
>
> We thank the reviewer for this important point. We agree that the original submission over-emphasized higher temperatures and did not sufficiently evaluate more conventional settings.
> Our initial focus on T $\in$ {1.0, 1.5, 2.0} was motivated by prior work, especially Top-H, and by the fact that differences between truncation-based logits processors tend to shrink as temperature decreases. Following the reviewer’s suggestion, we added experiments at T $\in$ {0.5, 0.7} on GSM8K, GPQA, and AlpacaEval for all three models and all four decoding methods (we were not able to complete MT-Bench in time).
>
>
> Please **refer to our response to Reviewer 2TUp** for detailed results reporting models performance in the tables for T=0.5 and T=0.7.
>
> These new results show a more nuanced picture. At T$\in${0.5,0.7}, Top-W no longer dominates nearly all (T,model) settings as it did in the original higher-temperature results, which is expected because truncation-based methods become more similar at low temperature. Encouragingly, Top-W remains strongest overall across the completed low-temperature reruns:  Min-p, Top-p, Top-H and Top-W win in 1,0,1,4 cases in GSM8K, 1,1,2,4 cases in GPQA, and 0,1,1,4 cases in AlpacaEval, respectively. We will include these results in the revised version of the paper.
>
>
> **Diversity:**
>
> We thank the reviewer for this suggestion. We would like to clarify that the paper already includes an explicit diversity evaluation in Appendix L. Following Top-H, our rubric-based LLM-as-a-judge setup scores five dimensions on a 1–10 scale, including diversity as one of them. In Appendix Tables 5, 6, and 7, we report the diversity-related score under column M1 for different models and temperatures. Across prompts, temperatures, and models, Top-W achieves the strongest average rubric score while remaining competitive on diversity, showing that its gains are not driven by sacrificing diversity relative to the baselines. We agree, however, that this was not emphasized clearly enough in the submission, and we will make this discussion more explicit in the revised version.
>
>
> **Whether Top-W is equivalent to other baselines at lower temperature:**
>
> Thank you for this question. We do not believe Top-W is simply equivalent to a baseline run at lower temperature. Lowering temperature only sharpens the model’s original next-token distribution before truncation. Top-W, by contrast, changes the truncation rule itself by selecting the kept set through an objective that combines geometric transport, entropy, and retained mass. This is also reflected in the theory (Section 4.3. in the paper). Under the uniform (0-1) metric, where transport ignores geometry, Top-W reduces to a probability-only criterion and recovers standard truncation rules such as Top-K and the Lagrangian form underlying Top-H.
>
> To make the role of geometry more explicit, we added a controlled ablation on AlpacaEval comparing Top-W under three ground metrics: a geometry-free uniform metric, an L1  metric, and the default diagonal Mahalanobis metric. The results show that geometry matters: the geometry-free control is weaker than both geometry-aware variants, and the diagonal Mahalanobis metric performs best overall. Please refer to our **response to Reviewer 2TUp** for detailed results of this experiment.
>
>
> **Reasoning-oriented benchmark/model:**
>
> We thank the reviewer for this suggestion. We agree that reasoning-oriented evaluation is important, and we therefore added results of Qwen3-4B-Thinking on AIME 2025:
>
> | | Min-p | Top-p | Top-H | Top-W|
> |---|---|---|---|---|
> | T=0.5 | 70.0 | 80.0 | 83.3 | 83.3 |
> | T=1.0 | 76.6 | 73.3 | 76.6 | 76.6 |
>
> The overall picture is that Top-W is on par with the baselines on this reasoning benchmark, with no large gap between methods. We note, however, that AIME contains only 30 questions, and we generated only one sample per question because the max-token budget is extremely large; thus, small score differences should not be over-interpreted. More broadly, reasoning benchmarks are a less controlled setting for decoder comparisons, since they require very long generations and substantially higher test-time compute, making it harder to isolate decoder-only effects. We will add this experiment in the revision.
>
>
> Thank you for taking the time to read and review our paper. We are grateful for your feedback, which substantially improved both the evaluation and presentation of the paper.

---

> > ### Author Rebuttal · Reviewer_XuZF · 2026-04-01
> >
> > Thank you for the response.
> >
> > My concerns related to diversity and evaluations were addressed. Therefore, I decide to raise the score.

---

### Official Review · Reviewer_2TUp · 2026-03-11

**Soundness:** 3
**Presentation:** 2
**Significance:** 2
**Originality:** 3
**Overall Recommendation:** 4
**Confidence:** 3

**Summary:**

The paper proposes Top-W, a new language model decoding algorithm. It frames decoding as an Optimal Transport problem and provides an efficient approximate solution method. It is similar in spirit to top-p or top-k; it constrains the number of tokens to be randomly sampled. However, Top-W adds other criteria into account, other than probability mass, mainly a notion of cost of going from a base token distribution to a reduced set based on the distances of the individual token embeddings. This aims to balance preserving probability mass over the selected tokens, a notion of transport cost from the original distribution based on token distances, and uncertainty through an entropy term.

The experiments show that for high-temperature regimes, the method outperforms many strong baselines. However, it remains to be shown if the method compares with the baselines on lower-temperature settings as well.

**Compliance With Llm Reviewing Policy:**

Affirmed.

**Final Justification:**

The author has addressed my concerns with the evaluation methodology, and the results are still convincing.

**Key Questions For Authors:**

- **Lines 93 to 95.** What do you mean by “9  (T,model) settings with $T \in \{1.0, 1.5, 2.0\}$  and 3 LLMs”? What is T here? This is not clear. Please explain more explicitly that  T refers to temperature, and clarify how these temperature values are used across the different model settings.

- **Lines 34 to 37, column 2.** You have the wrong citation here. The text refers to contrastive decoding, but the cited paper is about contrastive learning, which is a very different topic. I think the intended citation is: _Contrastive Decoding: Open-ended Text Generation as Optimization_ (Li et al., 2022).

- **Motivation for the OT-based truncation.** The paper provides end-task evaluation of the decoding method and a rigorous derivation, but it lacks evidence for why using OT in embedding space should help in practice/LM theory. Is there an ablation that could better convey the value of this selection process, specifically which tokens are included in the truncation set? That would help motivate the method more clearly ( this could definitely improve the work ). Any observed pattern in good quality text that you are capturing with the way you are doing distribution sharpening ?

* It is really important for you to include experiment with typical temperature ranges in the main experiments. Specially for the baselines. As it stands the evaluation does an unfair comparison of the baselines.

**Limitations:**

yes

**Strengths And Weaknesses:**

## Strengths

- The method is theoretically sound, novel, and computationally lightweight.
- For the high temperature settings, it seems to be better than many popular baselines.

## weaknesses

- **Temperature ranges used during evaluation.** I do not understand the choice of temperatures used in the main evaluation. In practice and research, people usually use temperatures below 1.0, yet the paper focuses on $T \in \{1.0, 1.5, 2.0\}$ . Why not also experiment with more conventional settings such as 0.4, 0.6, 0.8, 1.00.4,0.6,0.8,1.0?

  The decoding baselines you cite in the introduction and related work do not follow this practice. For instance, typical sampling and top-p papers experiment with temperature values smaller than 1.0. The top-H paper that you also reference does use 1, 1.5 and 2 but this is highly unconventional.

  Note that T=1 settings already seem to be the ones where Top-W has less of an edge. At higher temperatures, the baselines degrade dramatically. For example, Top-p appears to drop from 81 to 7 on Phi GSM8K, and from 71.27 on Qwen2.5-3B to 9. These temperature settings are not typically used in practice, and as far as I know, they are also not the standard settings used when evaluating other decoding algorithms. It is fine to use them with your method if the method specifically requires higher temperatures, but then the baselines should still be compared under fair and conventional conditions. **This aspect of your setup severely damages the evaluation of the method.**

- **Hyperparameter selection.** In Appendix M, you outline the grid of hyperparameters you search over and then select the best-performing configuration. It would be valuable to state clearly which dataset was used for this selection, to reassure the reader that the choice was not made based on any of the test sets reported in the evaluation.

---

> ### Author Rebuttal · Authors · 2026-03-31
>
> We sincerely thank the reviewer for the thoughtful feedback and for recognizing the method’s theoretical grounding, novelty, and efficiency.  We have made every effort to address the concerns. If the reviewer takes these points into consideration, we would be grateful if you could adjust your score in a positive way.
>
> **Temperature ranges:**
>
> We thank the reviewer for this important point. We agree that the original submission over-emphasized higher temperatures and did not sufficiently evaluate more conventional settings.
>
> Our initial focus on T $\in$ {1.0, 1.5, 2.0} was motivated by prior work, especially Top-H and Min-p, and by the fact that differences between truncation-based logits processors tend to shrink as temperature decreases. Per your suggestion we added the experiment for T $\in$ {0.5, 0.7} over GSM8K, GPQA and AlpacaEval, (we skipped MT-Bench as we couldn’t finish it in time) for all three backbone models and all four decoding methods.
>
>
> GSM8K:
>
> | T = 0.5 | Qwen | LLaMA | Phi |
> |---|---|---|---|
> | Min-p | **75.8** | 75.3 | 84.5 |
> | Top-p | 74.8 | 75.6 | 85.1 |
> | Top-H | 74.9 | 77.1 | 85.4 |
> | Top-W | 75.4 | **77.2** | **85.9** |
>
> | T = 0.7  | Qwen | LLaMA | Phi |
> |---|---|---|---|
> | Min-p | 74.5 | 73.1 | 83.6 |
> | Top-p | 75.7 | 73.1 | 84.7 |
> | Top-H | 74.4 | **77.6** | 84.6 |
> | Top-W | **75.8** | 77.1 | **85.4** |
>
>
> GPQA:
>
> | T = 0.5 | Qwen | LLaMA | Phi |
> |---|---|---|---|
> | Min-p | 29.2 | **31.9** | 30.9 |
> | Top-p | **30.5** | 29.2 | 31.8 |
> | Top-H | 27.6 | 30.3 | **32.1** |
> | Top-W | **30.5** | 30.8 | **32.1** |
>
> | T = 0.7  | Qwen | LLaMA | Phi |
> |---|---|---|---|
> | Min-p | 27.0 | 30.1 | 28.5 |
> | Top-p | 27.4 | 29.2 | 29.9 |
> | Top-H | 27.6 | **30.5** | 32.5 |
> | Top-W | **31.4** | 29.2 | **33.0** |
>
>
> AlpacaEval:
>
> | T = 0.5 | Qwen | LLaMA | Phi |
> |---|---|---|---|
> | Min-p | 10.7 | 26.9 | 21.7 |
> | Top-p | **11.1** | 26.9 | 21.2 |
> | Top-H | 10.2 | 26.4 | 23.1 |
> | Top-W | 10.0 | **27.0** | **24.8** |
>
> | T = 0.7 | Qwen | LLaMA | Phi |
> |---|---|---|---|
> | Min-p | 9.1 | 25.8 | 22.6 |
> | Top-p | 7.2 | 26.9 | 21.2 |
> | Top-H | 8.1 | **28.6** | 21.8 |
> | Top-W | **9.7** | 26.7 | **23.1** |
>
>
> These new results show a more nuanced picture. At T $\in$ {0.5, 0.7}, Top-W no longer dominates nearly all (T,model) settings as it did in the original higher-temperature results, which is expected because truncation-based methods become more similar at low temperature. Encouragingly, Top-W remains strongest overall across the completed low-temperature reruns: Min-p, Top-p, Top-H and Top-W win in 1,0,1,4 cases in GSM8K, 1,1,2,4 cases in GPQA and 0,1,1,4 cases in AlpacaEval, respectively. We will include these results in the revised version of the paper.
>
>
> **Hyperparameter selection:**
>
> We thank the reviewer for pointing this out. We agree that this was not stated clearly enough. In the revised version, we will explicitly clarify that hyperparameters were selected on a small dev set. Specifically, following Top-H, we held out 50 random examples from AlpacaEval as a development set and performed the $(\lambda,\beta)$ grid search only on that subset. After selecting the hyperparameters, we fixed them across all experiments.
>
> **OT-based truncation:**
>
> We thank the reviewer for this helpful suggestion. To make the role of geometry more explicit, we added a controlled ablation the same as the AlpacaEval experiment in the paper for phi-3-mini-4k-instruct at T $\in$ {0.5,1.0}. We compared Top-W under three ground metrics: a uniform / constant metric (geometry-free control), an L1 metric (simple geometry baseline), and the paper’s default diagonal Mahalanobis metric.
>
> | | Constant | L1 | Mahalanobis |
> |---|---|---|---|
> | T = 0.5 | 23.0 | 24.1 | 24.8 |
> | T = 1.0 | 21.1 | 21.9 | 22.4 |
>
> This result shows that the choice of geometry matters: the geometry-free control is weaker than both the L1 metric and the diagonal Mahalanobis metric, while the diagonal Mahalanobis metric is slightly better than the L1 metric. Overall, these results suggest that the gain comes from introducing meaningful token-space geometry, with the diagonal Mahalanobis metric giving the strongest performance. This supports our central claim that OT helps because it makes truncation sensitive to semantic proximity in embedding space rather than only to probability mass and entropy. We will add these results in the revised version of the paper.
>
>
> **Clarifying T and the citation:**
>
> We thank the reviewer for both points. In the revised version, we will make the notation more explicit by stating that T denotes the decoding temperature, so “9 (T,model) settings” means the 3 temperatures T $\in$ {1.0, 1.5, 2.0} evaluated for each of the 3 LLMs. We will also correct the citation in the contrastive decoding discussion. The intended reference was in fact Li et al. (2022).
>
> Thank you for taking the time to read and review our paper. We are grateful for your feedback, which substantially improved both the evaluation and presentation of the paper.

---

> > ### Author Rebuttal · Reviewer_2TUp · 2026-04-01
> >
> > I thank the authors for their response, and I have updated my score.

---

### Official Review · Reviewer_TxP4 · 2026-03-15

**Soundness:** 4
**Presentation:** 4
**Significance:** 3
**Originality:** 3
**Overall Recommendation:** 5
**Confidence:** 4

**Summary:**

This work proposes an sampling algorithm for language models in order to balance the trading of consistency and diversity. The proposed approach employs a Wasserstein distance to quantify the transport of the probability mass to avoid distortion, an entropy to encourage the more sharper distribution, and the penalty to  discourage discarding the probability mass. The algorithm leads to a two-step approach: f-step for selecting potentials for transporting probability mass given a current set, and s-step to update the current set using the potentials. Experiments show that proposed approach achieve gains on diverse benchmarks when compared with several sampling approaches and language models.

**Compliance With Llm Reviewing Policy:**

Affirmed.

**Final Justification:**

The rebuttal resolves my minor concern to this work.

**Key Questions For Authors:**

$S^c$ for Equation 6 is not explained clearly. I suspect it probably means a complement set. Otherwise, I might have misunderstood the discussion in section 3.

**Limitations:**

yes.

**Strengths And Weaknesses:**

Strengths
- This work proposes a novel algorithm for sampling based on Wasserstein distance with additional regularization terms. The approach is well motivated, carefully designed to avoid distortion of the transported probability mass.
- This paper is well written and easy to follow. The detail descriptions in section 3 are clearly presented, leading to the algorithm in section 4 with detail discussions on the choice of hyperparameters.
- Experiments were carried out on diverse models and task settings, demonstrating gains over other approaches.

Weaknesses
- The obvious weakness is a bit of increase in latencies as empirically demonstrated in section 6.2, but probably negligible given the gains when compared with other approaches.

---

> ### Author Rebuttal · Authors · 2026-03-31
>
> We sincerely thank the reviewer for the very positive assessment of the paper and for recognizing the novelty of the geometry-aware decoding objective, the clarity of the theory, and the breadth of the empirical evaluation.
>
> **$S^c$ for equation 6**:
>
> We thank the reviewer for pointing this out. You are correct: $S^c$ denotes the complement of the selected set S. To address this, we revised Lemma 3.1 to define  $S^c$ clearly and make the notation self-contained:
>
> Lemma 3.1: Let $S\subseteq V$ with $\Gamma_S\in(0,1)$ and define $S^c$ as its complement in $V$. Then
> $W_1(p,q_S) = (1-\Gamma_S) W_1(p(.|S^c),p(.|S))$,
>
> where $p(\cdot\mid S)$ and $p(\cdot\mid S^c)$ are the conditional distributions of $p$ restricted to $S$ and $S^c$, respectively.
>
> In the revision, we also add a short explanation after Eq. (6) emphasizing the interpretation of this factorization: $W_1(p,q_S)$ separates into (1) the amount of removed probability mass, $\(1-\Gamma_S\)$, and (2) the geometric distance between the removed and retained conditional distributions. We believe this clarification improves the clarity of the discussion in Section 3.
>
> Thank you so much for taking the time to read and review our paper. We are very grateful for the supportive and constructive review.

---

> > ### Author Rebuttal · Reviewer_TxP4 · 2026-04-03
> >
> > Thank you for the explanation, and it sounds good to me. I'll maintain my score.

---

### Decision · Program_Chairs · 2026-04-30

**Decision:**

Accept (spotlight)

**Comment:**

*Motivation:* Current token sampling (decoding) strategies in LLMs lack a principled framework for balancing faithfulness, creativity, and exploration for the text generation.

*Main Contribution:* The paper propose a novel sampling method grounded in optimal transport that formally characterizes and controls the trade-off between faithfulness, creativity, and exploration for sampling.

*Reviewer Summary:* Reviewers highlight the novelty of the approach, particularly its strong theoretical foundation, which translates into an effective and practical method. The primary concerns focus on the experimental evaluation, specifically the choice of temperature grid and the method’s stronger performance in high-temperature regimes.

*Rebuttal Summary:* The rebuttal successfully addresses the reviewers’ concerns, leading to increased confidence and improved scores across all reviews.

*Conclusion:* This work presents a mathematically grounded framework with significant practical implications. The Area Chair recommends acceptance.